# Few-Shot Adversarial Prompt Learning on Vision-Language Models

**Yiwei Zhou**
School of Automation
Beijing Institute of Technology
zhouyiwei@bit.edu.cn

**Xiaobo Xia**
Sydney AI Centre
University of Sydney
xiaoboxia.uni@gmail.com

**Zhiwei Lin**[*]
School of Automation
Beijing Institute of Technology
linzhiwei@bit.edu.cn

**Bo Han**
Department of Computer Science
Hong Kong Baptist University
bhanml@comp.hkbu.edu.hk

**Tongliang Liu**[*]
Sydney AI Centre
University of Sydney
tongliang.liu@sydney.edu.au

## Abstract

The vulnerability of deep neural networks to imperceptible adversarial perturbations has attracted widespread attention. Inspired by the success of vision-language foundation models, previous efforts achieved zero-shot adversarial robustness by aligning adversarial visual features with text supervision. However, in practice, they are still unsatisfactory due to several issues, including heavy adaptation cost, suboptimal text supervision, and uncontrolled natural generalization capacity. In this paper, to address these issues, we propose a few-shot adversarial prompt framework where adapting input sequences with limited data makes significant adversarial robustness improvement. Specifically, we achieve this by providing adversarially correlated text supervision that is end-to-end learned from adversarial examples. We also propose a novel training objective that enhances the consistency of multi-modal features while encourages differentiated uni-modal features between natural and adversarial examples. The proposed framework gives access to learn adversarial text supervision, which provides superior cross-modal adversarial alignment and matches state-of-the-art zero-shot adversarial robustness with only 1% training data. Code is available at: https://github.com/lionel-w2/FAP.

## 1   Introduction

The seminal works [1, 2] reveal that adversarial examples [2], consisting of malicious perturbations imperceptible to humans, can easily mislead state-of-the-art deep neural networks (DNNs) [3–6] into making incorrect predictions. This vulnerability limits the application of DNNs in safety-critical areas, such as medicine [7], healthcare [8], and autonomous driving [9].

Human cognition is immune to the distribution variations induced by adversarial attacks, reflecting a fundamental difference between human and machine cognitive understanding. Humans primarily

---

[*]Corresponding author: Tongliang Liu (tongliang.liu@sydney.edu.au) and Zhiwei Lin (linzhiwei@bit.edu.cn)

38th Conference on Neural Information Processing Systems (NeurIPS 2024).

rely on semantic information [10] from the context, while machines depend more on statistical distributional associations. Consequently, recent work [11] introduces text supervision in adversarial adaptation through foundational vision language models (VLMs) [12–19], enhancing adversarial robustness with improved semantic understanding. Specifically, they adapt visual prompts by aligning adversarial visual features with static text supervision from the CLIP model [12]. By narrowing the gap in the probability distribution between adversarial text-image logits and the ground-truth label, they achieve zero-shot adversarial robustness in downstream tasks.

However, although some progress has been made with the previous method, there are still three limitations to overcome before leveraging context to mitigate adversarial vulnerabilities. First, zero-shot adversarial robustness in downstream tasks stems from aligning image and text embeddings on large-scale generic datasets like the entire ImageNet [20] through adversarial adaptation, which necessitates a huge amount of time and computational resources. Second, static hand-crafted text prompts lack adversary-related hints, providing only content-related information while disregarding adversarial components. Finally, the current adaptation method only considers adversarial inputs while disregarding natural inputs. On the one hand, it fails to account for the relationship and distinctions between natural and adversarial examples, potentially leading to catastrophic forgetting of natural generalization during adversarial adaptation. Worse still, if there are distributional discrepancies in the downstream datasets, the constrained natural generalization could hinder the learning of robustness.

To address these issues, we propose a ***Few-shot Adversarial Prompt learning (FAP)*** framework where pre-trained VLMs are adversarially adapted in a few-shot manner [21, 22] with prompt learning [23–28]. This adapts the inputs rather than the parameters of the model. To the best of our knowledge, this is the first time to learn adversarial robustness from the perspective of few-shot prompt tuning. Due to the scarcity of data for establishing robust decision boundaries, the robust representations learned by existing adversarial visual prompt methods [11] are far from satisfactory. This leads us to rethink how to provide appropriate prompts for adversarial examples. Instead of using static hand-crafted text prompts, we propose to learn adversarially correlated text supervision end-to-end from adversarial examples. Moreover, we design a novel training objective that harmonizes the connection and distinction of natural and adversarial features from information across different modalities. That is, we force the multi-modal features of natural and adversarial inputs to be consistent while encouraging the differentiation between uni-modal embeddings.

Compared to existing methods, our method has several advantages. (1) It significantly reduces the dependence on abundant data, as both text supervision and learning objectives are adversarially correlated with visual embeddings, providing a better alignment to establish robust generalization from limited examples. By adapting with a 16-shot subset from ImageNet-1K, we achieve comparable zero-shot robustness in downstream tasks using only 1% training data. (2) We provide adversarially correlated text supervision learned end-to-end from adversarial examples, which notably improves the alignment between visual and textual embeddings, making superior zero-shot adversarial robustness. (3) Our novel training objective fully leverages the dual-encoder architectural advantage of CLIP. It enhances cross-modal consistency between natural and adversarial examples to avoid potential robustness generalization failures, while encourages uni-modal divergence to introduce an adversarial aware mechanism that aids in learning adversarial text supervision.

Before delving into details, we clearly summarize our contributions as follows. (1) We focus on a realistic and important research problem and discuss three major issues in previous adversarial prompt learning paradigms, potentially inspiring further improvements in this area. (2) To tackle these issues, we propose a novel adversarial few-shot prompt learning framework with learnable adversarial text supervision and an adversarial-aware prompt learning objective. This method is lightweight yet makes significant adversarial generalizations. (3) We justify our claims through a series of experiments on 11 benchmark datasets covering multiple recognition tasks. The proposed method significantly outperforms state-of-the-art adversarial prompt learning methods in adversarial few-shot learning, adversarial zero-shot transfer, and adversarial base-to-new generalization settings. Comprehensive ablation studies and discussions are also provided in Section 4.3 and Appendix D.

## 2   Preliminary

**CLIP recap.** A pre-trained CLIP model typically includes an image encoder $\mathcal{I}$ with learned parameters $\theta_{\mathcal{I}}$ and a text encoder $\mathcal{T}$ with learned parameters $\theta_{\mathcal{T}}$. Here we consider a $K$-class classification

problem for an image $\mathbf{x}$ and its corresponding label $y \in \{1, \ldots, K\}$. To perform zero-shot evaluation, $\mathbf{x}$ is first divided into $M$ patches and converted into the patch embeddings $e(\mathbf{x})$. A class token $c_{\text{cls}}$ is then appended to the patch sequence as $e(\mathbf{x}) = \{c_{\text{cls}}, e_1(\mathbf{x}), \ldots, e_M(\mathbf{x})\}$. Afterward, the image encoder $\mathcal{I}$ processes this embedded patch sequence with ViT [29] blocks to produce the latent image feature representation $\mathbf{z}^{(I)} = \mathcal{I}(e(\mathbf{x}); \boldsymbol{\theta}_{\mathcal{I}})$. For the text branch, we prepare hand-craft prompts $t_i \in \boldsymbol{t} = \{t_1, \ldots, t_K\}$ by appending the class name to a word template, such as 'a photo of a {class}'. Subsequently, $t_i$ is tokenized and embedded as $\boldsymbol{w}(t_i) = \{w_1(t_i), \ldots, w_N(t_i), i\}$, where $i$ corresponds the $i$-th class. The text encoder $\mathcal{T}$ then encodes these work embeddings into the latent text feature representation $\mathbf{z}^{(t_i)} = \mathcal{T}(\boldsymbol{w}(t_i); \boldsymbol{\theta}_{\mathcal{T}})$. For zero-shot classification, the probability of the image $\mathbf{x}$ in the $i$-th class is

$$p(y = i \mid \mathbf{x}) = \frac{\exp\left(\cos\left(\mathbf{z}^{(I)}, \mathbf{z}^{(t_i)}\right)/\tau\right)}{\sum_{j=1}^{K} \exp\left(\cos\left(\mathbf{z}^{(I)}, \mathbf{z}^{(t_j)}\right)/\tau\right)}, \tag{1}$$

where $\cos(\cdot, \cdot)$ denotes the cosine similarity score and $\tau$ is the temperature parameter.

**CLIP-based prompt learning.** Instead of adopting a hand-crafted prompt, prompt learning attempts to train lightweight learnable prompts $\boldsymbol{P_t}$ with a few examples from *downstream* data. To be concrete, $\boldsymbol{P_t}$ is inserted into word embeddings as $\boldsymbol{w}(t_i, \boldsymbol{P_t}) = \{\boldsymbol{P_t}, w_1(t_i), \ldots, w_N(t_i), i\}$. Then, the text feature representation is $\mathbf{z}^{(t_i, \boldsymbol{P_t})} = \mathcal{T}(\boldsymbol{w}(t_i, \boldsymbol{P_t}); \boldsymbol{\theta}_{\mathcal{T}})$. To preserve the alignment characteristics of the joint image-text feature space for zero-shot capabilities, CLIP-based prompt learning optimizes the prompt tokens by narrowing the gap in the distribution between text-image logits and the ground-truth label using cross-entropy:

$$\boldsymbol{P_t^*} = \arg\min_{\boldsymbol{P_t}} \mathbb{E}_{(\mathbf{x}, y)} \mathcal{L}_{\text{CE}}\left(\cos(\mathbf{z}^{(I)}, \mathbf{z}^{(t_i, \boldsymbol{P_t})}), y\right), \tag{2}$$

where $\cos(\mathbf{z}^{(I)}, \mathbf{z}^{(t_i, \boldsymbol{P_t})})$ corresponds the text-image logits. We suggest readers check Zhou et al. [28] for more details about CLIP-based prompt learning.

**Adversarial visual prompt.** Adversarial prompt learning optimizes prompt tokens through adversarial training, enhancing model robustness in a relatively small adaptation cost without altering the pre-trained model. Mao et al. [11] achieves this by adjusting the visual prompt of adversarial images in joint text-image feature space. Notably, owing to the application of text-image contrastive loss during the generation of adversarial examples, the adapted model reveals zero-shot adversarial robustness on downstream tasks. Formally, let $(\mathcal{X}, d_\infty)$ be the input feature space $\mathcal{X}$ with the infinity distance metric, where $d_\infty(\mathbf{x}, \mathbf{x}') = \|\mathbf{x} - \mathbf{x}'\|_\infty$. Adversarial data $\tilde{\mathbf{x}}$ falls in to close ball $\mathcal{B}_\epsilon(\mathbf{x})$ of radius $\epsilon$ centered at $\mathbf{x} \in \mathcal{X}$. That is, $\mathcal{B}_\epsilon(\mathbf{x}) = \{\mathbf{x}' \in \mathcal{X} \mid d_\infty(\mathbf{x}, \mathbf{x}') \leq \epsilon\}$. The learnable image prompt $\boldsymbol{P_v}$ is inserted to the visual patch embedding of $\tilde{\mathbf{x}}$, as $e(\tilde{\mathbf{x}}, \boldsymbol{P_v}) = \{c_{\text{cls}}, \boldsymbol{P_v}, e_1(\tilde{\mathbf{x}}), \ldots, e_M(\tilde{\mathbf{x}})\}$. Then, adversarial data $\tilde{\mathbf{x}}$ is generated by maximizing the text-image contrastive loss as

$$\tilde{\mathbf{x}} = \arg\max_{\tilde{\mathbf{x}} \in \mathcal{B}_\epsilon(\mathbf{x})} \mathcal{L}_{\text{CE}}\left(\cos(\tilde{\mathbf{z}}^{(I, \boldsymbol{P_v})}, \boldsymbol{t}), y\right), \tag{3}$$

where $\tilde{\mathbf{z}}^{(I, \boldsymbol{P_v})} = \mathcal{I}(e(\tilde{\mathbf{x}}, \boldsymbol{P_v}); \boldsymbol{\theta}_{\mathcal{I}})$. The learnable prompt token $\boldsymbol{P_v}$ is optimized given the adversarial example $\tilde{\mathbf{x}}$, hand-craft prompts $\boldsymbol{t}$, and ground-truth label $y$, by minimizing the adversarial text-image contrastive loss:

$$\boldsymbol{P_v^*} = \arg\min_{\boldsymbol{P_v}} \mathbb{E}_{(\mathbf{x}, y)} \mathcal{L}_{\text{CE}}\left(\cos(\tilde{\mathbf{z}}^{(I, \boldsymbol{P_v})}, \boldsymbol{t}), y\right). \tag{4}$$

Here, $\mathcal{L}_{\text{CE}}\left(\cos(\tilde{\mathbf{z}}^{(I, \boldsymbol{P_v})}, \boldsymbol{t}), y\right)$ is defined as a text-image contrastive adversarial training (TeCoA) loss by Mao et al. [11] that highlights adversarial text-image alignment.

**Drawbacks of previous methods.** Despite the promising zero-shot adversarial robustness achieved through adversarial visual prompts, certain inherent characteristics impede its widespread application.

(1) The zero-shot adversarial robustness in downstream tasks originates from the alignment of image and text embedding on a large-scale generic dataset like the entire ImageNet during prompt tuning. This necessitates an extensive amount of training data and employs prompts of considerable size (token-level prompts with a size of 200), which not only causes significant prompt-related overhead but also precludes the benefits of lightweight adaptation on the top of the pre-trained models that prompt tuning typically offers.

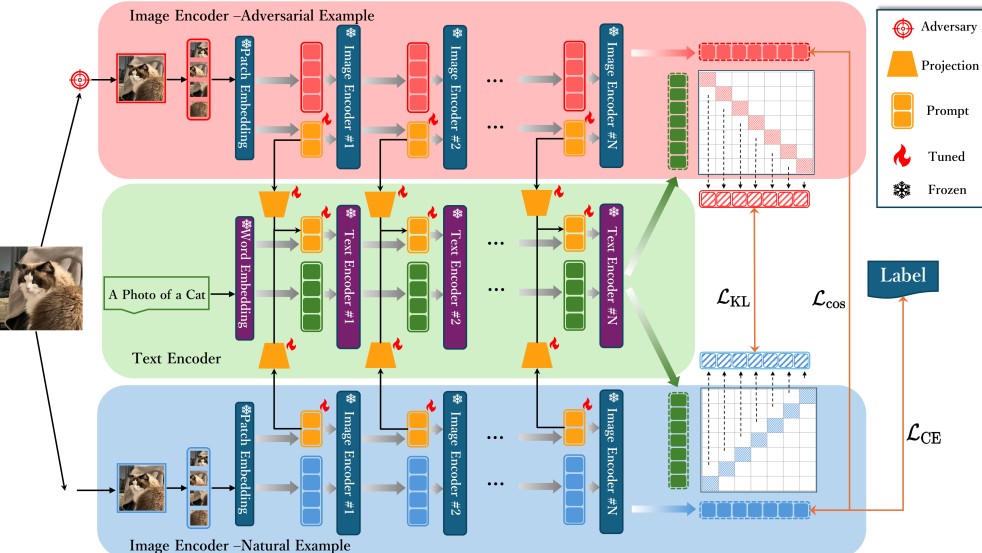

Figure 1: The overview of the proposed *Few-shot Adversarial Prompt learning (FAP)* framework. Note that only prompt tokens as well as the deep projections from image to text are tuned while the rest of the model is frozen. Our method promotes a consistent cross-modal similarity distribution between natural and adversarial examples, while encouraging differences in uni-modal representations. The adversarial-aware text supervision learned in this manner can better align adversarial features and establish robust decision boundaries with a limited number of examples. The natural and adversarial forward processes of the image encoder share parameters.

(2) Due to the distinct visual representation distribution between clean and adversarial examples, static hand-crafted prompts lack adversary-related hints, thereby only providing content-related information without effectively supervising the adversarial components contained in the images. However, manually adjusting hand-crafted prompts to inject additional adversarial hints is also challenging, as the imperceptibility of adversarial perturbations limits their feature description, and the intensity and distribution of these perturbations are variable throughout the training process.

(3) The current learning objective directly trains to provide prompts with adversarial examples, yet it overlooks the model capacity for natural generalization in downstream tasks. This presents a potential risk of failure, especially in the context of few-shot prompt tuning where the pre-trained model shows inadequate natural generalization on a sampled few-shot dataset.

## 3   Method

**Overview.** To address the limitations of previous methods, we propose FAP, a few-shot adversarial prompt learning framework. Our framework uses lightweight learnable prompts on the top of the pre-trained CLIP in a few-shot manner, as the case in natural prompt tuning [28]. In more detail, we introduce learnable prompt tokens for adversarial examples, which allows the model to provide more appropriate text supervision that helps balance natural and adversarial generalization. Based on CLIP's dual-encoder architecture, we further provide a novel training objective that guides the discrimination of natural and adversarial embeddings in uni-modal feature space. This promotes uni-modal divergence to incorporate an adversarial-aware mechanism, facilitating the learning of adversarial text supervision. The overview of the proposed framework is provided in Figure 1. Below, we discuss the FAP framework step by step.

### 3.1   Learnable Text Supervision for Adversarial Examples

When adapting the CLIP model, a slight change in wording could have a huge impact on performance [28]. With the existence of adversarial examples, the situation has become worse. The distribution differences between natural and adversarial examples necessitate the design of special-

ized text supervision specifically for adversarial samples. Therefore, we introduce text prompt tokens that are end-to-end learned through adversarial examples.

Formally, our adversarial prompt learning is implemented on a few-shot subset $\mathcal{S}$, created by sampling $m$ examples from each of the $K$ classes in the original dataset. Learnable prompts consist of both visual and text branches, denoted as $\boldsymbol{P} = \{\boldsymbol{P_v}, \boldsymbol{P_t}\}$. The visual prompt token $\boldsymbol{P_v}$ is incorporated into the image embedding, as observed in an adversarial visual prompt, while text prompt token $\boldsymbol{P_t}$ is inserted into word embedding, as is the case in natural prompt learning. To preserve mutual synergy between visual and text branchs, $\boldsymbol{P_t}$ is obtained from $\boldsymbol{P_v}$ through linear projection $h$, which can be denoted as $\boldsymbol{P_t} = h\left(\boldsymbol{P_v}\right)$. The proposed framework can be categorized as a cross-modal prompt [30] with minimal modification for adversarial robustness tasks. We offer a comprehensive analysis of the prompt design in Section 4.3.

## 3.2 Balancing Natural and Adversarial Generalization in Few-Shot Adversarial Prompt

For adapting the CLIP model to adversarial robustness tasks, the existing method [11] proposes the TeCoA loss (Eq.(4)). This method minimizes the discrepancy between the distribution of adversarial text-image similarity and one-hot ground-truth labels. While this strategy effectively aligns text representations during adversarial adaptation, it potentially compromises the model's generalization ability in specific recognition tasks under few-shot conditions.

The method's effectiveness depends on the similarity between the downstream task's distribution and the pre-trained representations. When the downstream task closely aligns with the pre-trained representation, the CLIP model shows preferable natural generalization, and adding learnable prompts for robustness adaptation is advantageous. However, a significant mismatch between the downstream distribution and pre-trained representations challenges the CLIP model's natural generalization capabilities. In such cases, expecting prompt tokens to learn both natural and robust generalization from a few adversarial examples is overly ambitious.

**Balancing natural and adversarial generalization.** Inspired by the success of TRADES [31] in standard adversarial training, we propose a surrogate adversarial text-image contrastive loss that decouples the adversarial text-image contrastive loss into natural and adversarial terms. By encoding image and text embeddings with their respective transformer encoder and calculating similarity across modality, we have the natural and adversarial text-image logits: $\cos(\mathbf{z}^{(I,\boldsymbol{P_v})}, \mathbf{z}^{(\boldsymbol{t},\boldsymbol{P_t})})$ and $\cos(\tilde{\mathbf{z}}^{(I,\boldsymbol{P_v})}, \mathbf{z}^{(\boldsymbol{t},\boldsymbol{P_t})})$, where $\mathbf{z}^{(\boldsymbol{t},\boldsymbol{P_t})} = \{\mathbf{z}^{(t_1,\boldsymbol{P_t})}), \ldots, \mathbf{z}^{(t_K,\boldsymbol{P_t})})\}$. The learning objective can be stated as:

$$\mathcal{L} = \mathcal{L}_{\text{CE}}\left(\cos(\mathbf{z}^{(I,\boldsymbol{P_v})}, \mathbf{z}^{(\boldsymbol{t},\boldsymbol{P_t})}), y\right) + \lambda \mathcal{L}_{\text{KL}}\left(\cos(\mathbf{z}^{(I,\boldsymbol{P_v})}, \mathbf{z}^{(\boldsymbol{t},\boldsymbol{P_t})}), \cos(\tilde{\mathbf{z}}^{(I,\boldsymbol{P_v})}, \mathbf{z}^{(\boldsymbol{t},\boldsymbol{P_t})})\right), \quad (5)$$

where $\mathcal{L}_{\text{KL}}$ denotes the Kullback–Leibler (KL) divergence and $\lambda$ is a weight parameter. In Eq. (5), the first term encourages minimizing the natural error between the natural text-image similarity and label. The second term minimizes the boundary error by narrowing the distribution gap between natural and adversarial text-image similarity to ensure cross-modal adversarial consistency. Note that a balanced two-term objective is crucial for downstream generalization, as this design alleviates the potential failure in robustness caused by discrepancies in natural generalization. We provide more analysis on the natural generalization gap in Appendix D.2.

## 3.3 Uni-Modal Adversarial-Aware Mechanism

To fully leverage the structural advantages of CLIP, we go beyond enforcing consistency constraints on cross-modal text-image features and tailor adversarial robustness enhancements for uni-modal features. Specifically, we introduce an adversarial-aware mechanism for visual features, guiding the distinction between natural and adversarial examples. To the best of our knowledge, this is the first initiative to foster differentiated representations in adversarial regularization.

Given the distinct distributions of natural and adversarial examples, we argue that driving consistent outputs for natural and adversarial examples in visual models constitutes a compromise, trading off generalization for robustness. In contrast, within CLIP, we achieve robustness by maintaining adversarial consistency in the text-image joint space with the adversarial term in Eq. (5), while preserving the distributional differences of features in the uni-modal visual space to minimize the

impact on generalization performance. Here, we append an extra constraint on the adversarial term with cosine similarity:

$$\mathcal{L}_{\cos} = \cos\left(\mathbf{z}^{(I,\boldsymbol{P_v})}, \tilde{\mathbf{z}}^{(I,\boldsymbol{P_v})}\right) + 1, \tag{6}$$

where the constant 1 maintains the *non-negativity* of $\mathcal{L}_{\cos}$. We introduce the adversarial-aware mechanism by adjusting prompt tokens to minimize similarity, thereby distinctly differentiating between natural and adversarial visual features. During the training process, the text branch learns to provide proper text supervision for different visual features, ensuring that the outputs in the text-image joint space are consistent for natural and adversarial embeddings, which have significant distributional differences in the visual space.

### 3.4 Overall Learning Objective

**Objective for outer minimization.** The overall training objective can be obtained by introducing uni-modal adversarial aware mechanism $\mathcal{L}_{\cos}$ to Eq. (5) as:

$$\mathcal{L}_{\text{final}} = \mathcal{L}_{\text{CE}}\left(\cos(\mathbf{z}^{(I,\boldsymbol{P_v})}, \mathbf{z}^{(t,\boldsymbol{P_t})}), y\right) + \lambda\mathcal{L}_{\cos} \cdot \mathcal{L}_{\text{KL}}\left(\cos(\mathbf{z}^{(I,\boldsymbol{P_v})}, \mathbf{z}^{(t,\boldsymbol{P_t})}), \cos(\tilde{\mathbf{z}}^{(I,\boldsymbol{P_v})}, \mathbf{z}^{(t,\boldsymbol{P_t})})\right). \tag{7}$$

**Objective for inner maximization.** The goal of inner maximization is to generate the adversarial example $\tilde{\mathbf{x}}$. Here, we leverage the adversarial term in Eq. (5) as this surrogate loss and find the adversarial example $\tilde{\mathbf{x}}$ as follows:

$$\tilde{\mathbf{x}} = \arg\max_{\tilde{\mathbf{x}} \in \mathcal{B}_\epsilon(\mathbf{x})} \mathcal{L}_{\text{KL}}\left(\cos(\mathbf{z}^{(I,\boldsymbol{P_v})}, \mathbf{z}^{(t,\boldsymbol{P_t})}), \cos(\tilde{\mathbf{z}}^{(I,\boldsymbol{P_v})}, \mathbf{z}^{(t,\boldsymbol{P_t})})\right). \tag{8}$$

Note that strong attacks can help robustness. Here, the general PGD attack formulation with the CE loss like Eq. (3) is also applicable. With the learning objective outlined in Eq. (7), we adapt learnable prompt $\boldsymbol{P} = \{\boldsymbol{P_v}, \boldsymbol{P_t}\}$ tokens on the few-shot dataset $\mathcal{S}$ as:

$$\boldsymbol{P}^* = \arg\min_{\boldsymbol{P}} \mathbb{E}_{(\mathbf{x},y)\sim\mathcal{S}} \mathcal{L}_{\text{final}}. \tag{9}$$

### 3.5 Intuition behind Objective Design

Our learning objective highlights the differentiated processing of features under different modalities, in which we introduce an additional adversarial-aware mechanism with uni-modal image features. We discuss the intuition behind the design concept. We visualize the uni-modal embedding to demonstrate the impact of the adversarial-aware mechanism on the model's feature learning.

In Figure 2a, we find that certain adversarial embeddings closely resemble natural examples. This suggests that the consistency of cross-modal features between natural and adversarial examples arises from the model's tendency to minimize loss by generating minimal adversarial perturbations. These exceedingly small perturbations do not effectively promote robust learning. In contrast, the adversarial-aware mechanism clearly separates the natural and adversarial embeddings in Figure 2b, preventing the minimal perturbation shortcut and guiding the model to recognize the differences between natural and adversarial image embeddings.

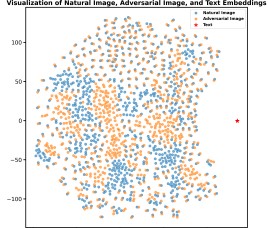
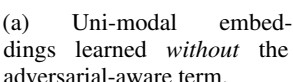
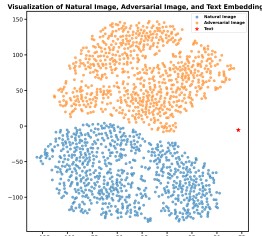

(a) Uni-modal embeddings learned *without* the adversarial-aware term.

(b) Uni-modal embeddings learned *with* the adversarial-aware term.

Figure 2: Visualization of the natural image embedding, adversarial image embedding, and text embedding after tuning with and without the adversarial-aware term. Images are sampled from the same class in the Caltech101 dataset [32].

For better understanding, we discuss different training objective designs and their results in Section 4.3 and describe our adversarial prompt learning and adversarial prompt testing pipeline in Appendix A. Additionally, we demonstrate the significant robustness gains our learning objective brings to other prompt designs through a case study. More details can be checked in Appendix D.4.

# 4 Experiments

## 4.1 Setups

**Baselines.** To demonstrate the expertise of the proposed method, we employ the adversarial version of multiple commonly used prompt learning designs for comparison. We categorize our baselines into two groups: (1) Methods using hand-crafted text supervision, such as zero-shot CLIP [12] and **AdvVP** [11]. (2) Methods utilizing learnable text prompts, including **AdvVLP** and **AdvMaPLe** [30]. Note that we primarily focus on learnable prompts extending the **AdvVP** framework. Details on pure text prompt effects in adversarial settings (**AdvTP**) [28] are discussed in Appendix D.11. Additional information about these methods and static prompt templates for each dataset are provided in Appendices C.1 and C.2, respectively.

**Datasets.** To evaluate the proposed method, we align with previous works [28, 33] and utilize 11 diverse image recognition datasets that span multiple vision tasks. Specifically, the datasets include two generic object datasets: ImageNet-1K [20] and Caltech101 [32]; a texture recognition dataset: DTD [34]; five fine-grained object recognition datasets: FGVCAircraft [35], OxfordPets [36], Flowers102 [37], Food101 [38], and StanfordCars [39]; a scene recognition dataset: SUN397 [40]; an action recognition dataset: UCF101 [41]; and a satellite image classification dataset: EuroSAT [42].

**Implementation details.** We conduct experiments on the ViT-B/32 CLIP architecture and report the average results over three random seeds. All models are trained for 5 epochs in cross-dataset evaluation and 10 epochs for other benchmark settings by using an SGD optimizer with a momentum of 0.9. The initial learning rate is set at 0.0035. We apply a cosine learning rate scheduler and a warm-up strategy during the first epoch. For adversarial prompt learning, we use token prompts of size 2 in both the vision and text branches across the first 9 transformer blocks. Attacks are generated under $\ell_\infty$ threat model through a 2-step PGD attack, with a perturbation boundary $\epsilon = 1/255$ and a step size $\alpha = 1/255$, following the methodologies outlined in [11]. The adversarial robustness is evaluated using a 100-step PGD attack.

Note that due to the limited space of the main paper, we provide comprehensive evaluations, including cross-dataset evaluation (Appendix D.1), the comparison with AdvMaPLe (Appendix D.3), alternative CLIP architectures (Appendix D.5), different attack strengths (Appendix D.6), various choices of adversarial robustness evaluation methods (Appendix D.7), and different training-time attack generation (Appendix D.8).

## 4.2 Main Results

**Adversarial few-shot learning.** In this scenario, we evaluate the model's ability to develop robust representations with a severely limited amount of downstream data. Specifically, we tune the model using $\{1, 2, 4, 8, 16\}$ shots from each class. As shown in Figure 3, the static text prompt of baseline method struggles to align with adversarial input images under a few-shot setting. Even with an increased number of training samples, the model's performance fails to improve, indicating difficulties in adversarial learning. AdvVLP and AdvMaPLe, through end-to-end learning of adversarial text prompt tokens from adversarial examples, have acquired the capability to adjust prompts from limited samples to gain adversarial robustness. By further training with our proposed objective, our method achieves superior average natural and adversarial accuracy across 11 datasets.

**Adversarial base-to-new generalization.** We present a more challenging adversarial base-to-new generalization setting, where datasets are bifurcated into base and new subclasses. Here, models are trained with a 16-shot dataset from the base classes and are subsequently evaluated on both base and new classes. In this setting, as the number of categories in datasets is generally much smaller than the number of examples per class, models need to learn intrinsic features within each dataset and robust representations from limited examples to effectively generalize large amounts of test data.

From Table 1, we observe that our method not only surpasses all its counterparts in robust metrics, but also reveals superior natural generalization due to the joint consideration of natural and robust features in our training objective. Additionally, our method also reveals much better stability (lower standard deviation). That is, even sampled few-shot subset has a natural generalization gap, our learning objective still works well and prevents potential failure.

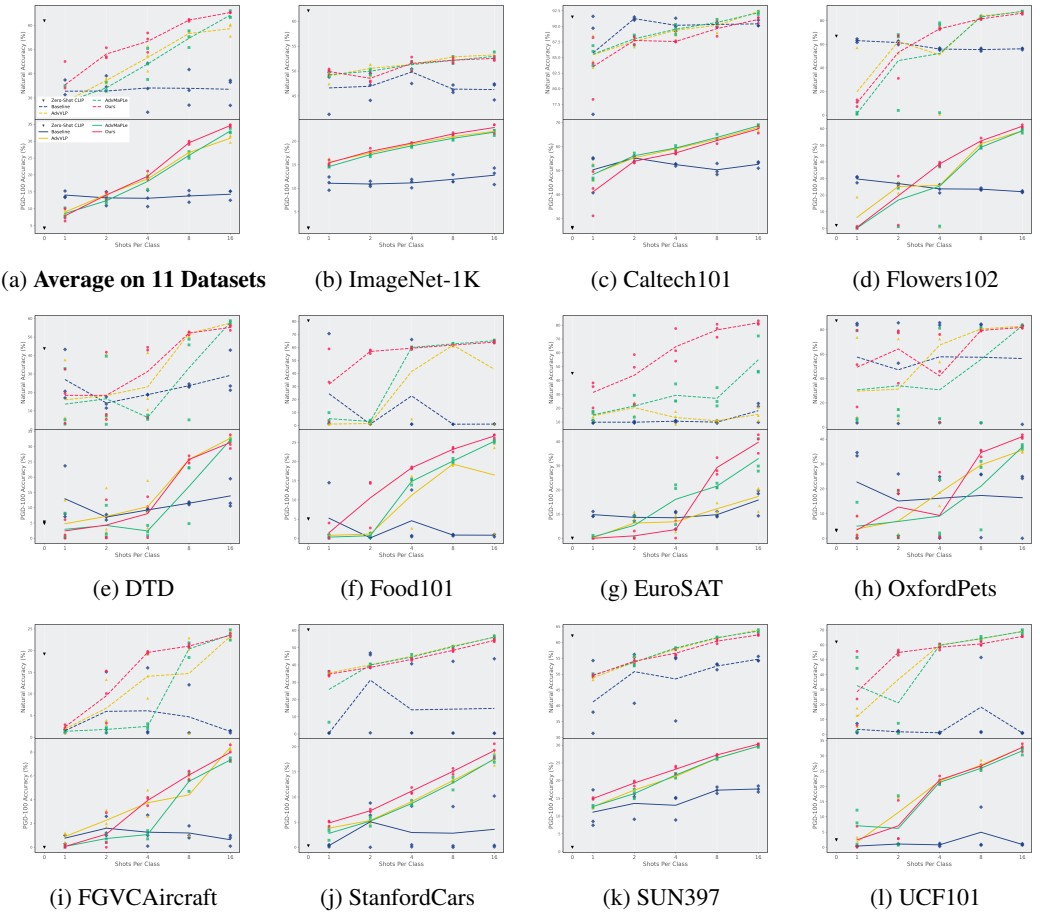

| | (a) Average on 11 Datasets | (b) ImageNet-1K | (c) Caltech101 | (d) Flowers102 |
| | (e) DTD | (f) Food101 | (g) EuroSAT | (h) OxfordPets |
| | (i) FGVCAircraft | (j) StanfordCars | (k) SUN397 | (l) UCF101 |

Figure 3: Accuracy (%) of adversarial few-shot learning on 11 datasets. The dots represent the result of each experiment and lines reveal the trend of the average results from three trials under each setting with respect to the shot numbers. In each subfigure, we report the natural accuracy (dashed line) in the upper half, and the robust accuracy (solid line) in the lower half. Statistical results of standard deviations across multiple trials are included in Appendix D.9.

Table 1: Adversarial base-to-new Generalization performance. We report the average result of the Base Natural Accuracy (%), Base Adversarial Accuracy (%), New Natural Accuracy (%), and New Adversarial Accuracy (%) on 11 datasets. Detailed results for each dataset are provided in Appendix D.10.

| Method | Base Class | | New Class | |
| | Base Nat Acc | Base Adv Acc | New Nat Acc | New Adv Acc |
|---|---|---|---|---|
| AdvVP | 31.68±6.57 | 14.43±2.26 | 30.39±6.40 | 13.36±2.80 |
| AdvVLP | 58.95±11.67 | 32.37±6.67 | 46.92±7.41 | 21.61±3.86 |
| AdvMaPLe | 60.38±8.03 | 30.69±4.71 | 46.18±6.39 | 20.25±3.39 |
| **FAP** | **70.52±0.82** | **38.05±2.15** | **49.58±3.55** | **21.86±2.57** |

**Matching Benchmark Zero-Shot Results Adapted with ImageNet-1K.** In addition to comparing with the baseline AdvVP under few-shot settings, we also benchmark against zero-shot results, where robustness is evaluated through cross-dataset evaluations. Initially adapted on ImageNet-1K, our method does not require adaptation across the entire dataset nor extensive prompt designs like the AdvVP [11], which uses embedding-level token prompts of size 200 and pixel-level pad prompters of size 40. As shown in Table 2, our method aligns with benchmark performance using just 1.25% of ImageNet-1K examples, significantly accelerating the training process by over 97%. Moreover, enhancements from 16-shot to 32-shot training and deepening prompt layers from 9 to 12 allow our method to exceed previous adversarial prompt tuning results.

Table 2: Comparison with benchmark result [11] which adapts models on the entire ImageNet-1K. We report the average natural and robust accuracy across downstream datasets. Running time is computed on a single NVIDIA RTX A40 GPU.

| Method | Dataset | Params (/M) | Time (/Day) | Average on Downstream Dataset | |
|---|---|---|---|---|---|
| | | | | Natural Acc (%) | PGD-100 Acc (%) |
| AdvVP | 16-shot (1.25%) | 0.07 | 0.65 | 41.96 | 12.97 |
| AdvVP | Entire (100%) | 0.24 | 49.9 | 46.58 | 25.21 |
| **FAP** | 16-shot (1.25%) | 0.42 | 0.71 | 48.18 | 25.06 |
| **FAP** | 32-shot (2.49%) | 0.43 | 1.43 | **49.93** | **25.39** |

## 4.3 More Analysis

**Trade-off between natural and adversarial robustness.** Aligning with the decoupled form of classical adversarial training [31], our prompt objective incorporates two terms that ensure the generalization of natural examples and the consistency of robust representations. This motivates us to investigate the trade-off between natural and adversarial robustness, and to dynamically adjust this trade-off depending on the desired level of adversarial robustness.

From Table 3, we can conclude that as $\lambda$ increases, the proportion of the adversarial component in the total loss increases, and the natural accuracy declines continuously. Meanwhile, adversarial robustness gradually improves, reflecting the trade-off between natural and adversarial generalization. However, when $\lambda$ becomes too large ($\lambda > 2.5$), continuing to increase the proportion of the adversarial component does not lead to further improvements in robustness.

Table 3: Adversarial base-to-new generalization performance (%) w.r.t. different $\lambda$ values.

| $\lambda$ | Base Class | | New Class | |
|---|---|---|---|---|
| | Base Nat Acc | Base Adv Acc | New Nat Acc | New Adv Acc |
| 1.0 | **71.95** | 36.31 | **52.47** | 22.34 |
| 1.5 | 70.60 | 39.15 | 51.79 | 23.65 |
| 2.0 | 68.46 | 40.36 | 46.99 | 23.73 |
| 2.5 | 68.44 | **41.38** | 48.49 | **23.90** |
| 3.0 | 67.15 | 40.58 | 46.15 | 22.84 |
| 3.5 | 66.49 | 39.04 | 41.57 | 20.64 |

**Prompt depth and prompt length.** We provide architectural ablation results for prompt design concerning different prompt depth and length settings. In Table 4, we can observe that increasing both prompt depth and prompt length introduces more learnable parameters, thereby resulting in improved performance. Furthermore, we can also conclude that the performance gain obtained by increasing prompt depth is higher than that achieved by increasing prompt length, and the improvement in robustness metric is larger than in natural accuracy.

**Ablation for training objective design.** In Section 3.4, we present our proposed novel training objective tailored for adversarial prompt learning. Our loss follows a two-term design, comprising a natural term and an adversarial term. The adversarial term further considers both the consistency and diversity of natural and adversarial features. In practice, we use KL divergence to constrain cross-modal consistency and encourage uni-modal diversity with cosine similarity. In Table 5, we present other possible designs for the loss function and conduct an ablation study under the adversarial base-to-new setting. Our method provides the best robustness across all these loss function settings.

Table 4: Natural and robust performance (%) w.r.t. different prompt depth and length settings. Results are obtained in under 16-shot adversarial prompt learning on StanfordCars.

| Nums | Prompt Depth | | Prompt Length | |
|---|---|---|---|---|
| | Natural Acc | PGD-100 Acc | Natural Acc | PGD-100 Acc |
| 2 | 71.60 | 19.00 | 82.60 | 56.90 |
| 4 | 75.50 | 41.50 | 85.30 | 59.20 |
| 6 | 77.50 | 49.50 | 84.40 | 61.10 |
| 8 | 80.10 | 52.80 | 84.00 | 60.00 |
| 10 | 82.20 | **58.00** | 84.90 | 60.00 |
| 12 | **84.00** | 57.30 | **85.50** | **61.80** |

**Instability analysis for deep prompt interaction.** We report an instability of generalization performance caused by the improper deep prompt interaction, revealing that the standard cross-modal prompt interaction design, from text to image prompt token, is not plug-and-play under the setting of adversarial robustness. When natural and adversarial terms are present in a certain moderate ratio in the learning objective, the performance of the model may experience a significant decline. From Figure 4, we find that the instability intensity caused by the text-to-image design varies across different datasets, and the values of $\lambda$ leading to this instability are also different. For instance, on some generic datasets, the performance degradation it usually brings is not significant (Figure 4c). However, on some fine-grained datasets, the significant performance degradation caused by this instability is unacceptable (Figure 4b).

Table 5: Ablation study of base-to-new generalization performance (%) w.r.t. different training objective design. Here, TeCoA, JS, KL, MAE, MSE and Cos stand for Text-image Contrastive Loss, Jensen-Shannon Divergence, Kullback-Leibler Divergence, Mean Absolute Error, Mean Squared Error and Cosine Similarity, respectively.

| Natural term | Adversarial term | | Base Nat Acc | Base Adv Acc | New Nat Acc | New Adv Acc |
| | Consistency | Diversity | | | | |
| --- | --- | --- | --- | --- | --- | --- |
| ✗ | TeCoA | ✗ | 57.96 | 30.10 | 43.73 | 19.01 |
| ✔ | TeCoA | ✗ | 48.18 | 26.57 | 36.52 | 16.41 |
| ✔ | JS | ✗ | 74.02 | 34.38 | 56.91 | 20.75 |
| ✔ | KL | ✗ | 71.20 | 37.70 | 49.52 | 21.18 |
| ✔ | KL | MSE | **77.73** | 20.34 | **64.73** | 15.90 |
| ✔ | KL | MAE | 74.02 | 30.56 | 57.41 | 17.59 |
| ✔ | KL | Cos | 70.60 | **39.15** | 51.79 | **23.65** |

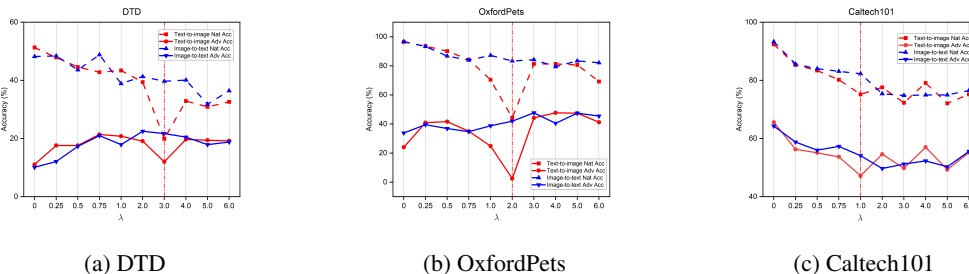

(a) DTD         (b) OxfordPets         (c) Caltech101

Figure 4: Instability analysis for DTD, OxfordPets, and Caltech101. We report the model performance (%) w.r.t the ratio ($\lambda$) between natural and robust terms in training objectives. The results of deep prompt interaction from text to image are plotted in red line, while that from image to text are plotted in blue line.

To understand this, we plot the loss curve during the training process under both stable and unstable settings. As revealed in Figure 5, in unstable cases, we observe that the robust loss drops to zero early in training and remains nearly unchanged at this low level during the mid-phase, while the overall loss does not decrease as expected. This suggests the text prompt falls into a trivial local solution during optimization, equating natural and adversarial logits. This nullifies the adversarial term but overlooks natural generalization, causing consistently high natural loss. This issue typically occurs when the natural and robust terms are balanced in a moderate ratio in the training objective.

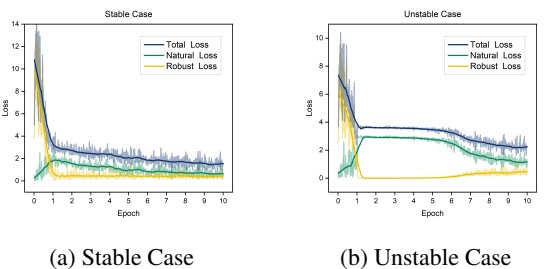

(a) Stable Case      (b) Unstable Case

Figure 5: Training loss curve under both stable and unstable settings. We report the total, natural, and robust loss during the whole training stage.

We propose a minimal refinement to prevent instability: switching the deep prompt interaction to an image-to-text scenario. Here, the text prompt is derived from the image prompt projection, limiting its adaptability. This prevents the adversarial loss from reaching zero, thus avoiding the issue.

## 5 Conclusion

In this paper, we focus on adversarial prompt tuning on vision-language models, a domain with significant potential for zero-shot downstream adversarial robustness. We precisely reveal the issues of previous methods that perform adversarial visual prompts with static text supervision. Our method distinguishes itself by introducing learnable adversarial text supervision combined with a new training objective, facilitating effective learning in a few-shot setting. The proposed method enjoys excellent algorithmic properties and matches state-of-the-art performance, notably with reduced computational demand. We believe that this work can provide some insights to the community and stimulate further research in this area.

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

# Appendix

# A  Pipelines of Adversarial Prompt Learning and Testing

For a better understanding of the designed algorithm, we describe our adversarial prompt learning and adversarial prompt testing pipeline in Algorithm 1 and Algorithm 2 respectively.

---

**Algorithm 1** Few-shot Adversarial Prompt Learning (FAP)

---

**Input:** The few-shot dataset $\mathcal{S}$, CLIP pre-trained model $\boldsymbol{\theta} = \{\boldsymbol{\theta}_{\mathcal{I}}, \boldsymbol{\theta}_{\mathcal{T}}\}$, prompt vectors $\boldsymbol{P} = \{\boldsymbol{P_v}, \boldsymbol{P_t} = h\,(\boldsymbol{P_v})\}$, text description $\boldsymbol{t}$, and weight parameter $\lambda$.
**for all** training epochs **do**
  **for all** $\mathbf{x}, y \in$ a minibatch **do**
    *# Calculate image and word embeddings*
    $\boldsymbol{e}(\mathbf{x}, \boldsymbol{P_v}) \leftarrow \{c_{\text{cls}}, \boldsymbol{P_v}, e_1(\mathbf{x}), \ldots, e_M(\mathbf{x})\}$;
    $\boldsymbol{w}(t_i, \boldsymbol{P_t}) \leftarrow \{\boldsymbol{P_t}, w_1(t_i), \ldots, w_N(t_i), i\}$;
    *# Generate clean visual and text representations*
    $\mathbf{z}^{(I, \boldsymbol{P_v})} \leftarrow \mathcal{I}(\boldsymbol{e}(\mathbf{x}, \boldsymbol{P_v}); \boldsymbol{\theta}_{\mathcal{I}})$;
    $\mathbf{z}^{(t_i, \boldsymbol{P_t})} \leftarrow \mathcal{T}(\boldsymbol{w}(t_i, \boldsymbol{P_t}); \boldsymbol{\theta}_{\mathcal{T}})$;
    *# Generate adversarial examples*
    $\tilde{\mathbf{x}} = \arg\max_{\tilde{\mathbf{x}} \in \mathcal{B}_\epsilon(\mathbf{x})} \mathcal{L}_{\text{KL}}\left(\cos(\mathbf{z}^{(I, \boldsymbol{P_v})}, \mathbf{z}^{(t, \boldsymbol{P_t})}), \cos(\tilde{\mathbf{z}}^{(I, \boldsymbol{P_v})}, \mathbf{z}^{(t, \boldsymbol{P_t})})\right)$;
    *# Compute the overall loss*
    $\mathcal{L}_{\text{final}} = \mathcal{L}_{\text{CE}}\left(\cos(\mathbf{z}^{(I, \boldsymbol{P_v})}, \mathbf{z}^{(t, \boldsymbol{P_t})}), y\right) + \lambda \mathcal{L}_{\text{cos}} \cdot \mathcal{L}_{\text{KL}}\left(\cos(\mathbf{z}^{(I, \boldsymbol{P_v})}, \mathbf{z}^{(t, \boldsymbol{P_t})}), \cos(\tilde{\mathbf{z}}^{(I, \boldsymbol{P_v})}, \mathbf{z}^{(t, \boldsymbol{P_t})})\right)$;

    *# Update prompt vectors*
    $\boldsymbol{P} \leftarrow \boldsymbol{P} - \nabla_{\boldsymbol{P}} \mathcal{L}_{\text{final}}$.
  **end for**
**end for**

---

---

**Algorithm 2** Adversarial Prompt Testing

---

**Input:** The test dataset $\mathcal{S}_{\text{test}} = \{(\mathbf{x}_i, y_i)\}_{i=1}^{n}$, CLIP pre-trained model $\boldsymbol{\theta} = \{\boldsymbol{\theta}_{\mathcal{I}}, \boldsymbol{\theta}_{\mathcal{T}}\}$, adapted prompt vectors $\boldsymbol{P}^* = \{\boldsymbol{P_v^*}, \boldsymbol{P_t^*}\}$, and text description $\boldsymbol{t}$.
**Output:** Natural accuracy `nat_acc`, adversarial accuracy `adv_acc`.
**Initialize:** `nat_correct` $\leftarrow 0$, `adv_correct` $\leftarrow 0$;
**for all** $\mathbf{x}, y \in \mathcal{S}_{\text{test}}$ **do**
  *# Calculate image and word embeddings*
  $\boldsymbol{e}(\mathbf{x}, \boldsymbol{P_v^*}) \leftarrow \{c_{\text{cls}}, \boldsymbol{P_v^*}, e_1(\mathbf{x}), \ldots, e_M(\mathbf{x})\}$;
  $\boldsymbol{w}(t_i, \boldsymbol{P_t^*}) \leftarrow \{\boldsymbol{P_t^*}, w_1(t_i), \ldots, w_N(t_i), i\}$;
  *# Generate clean visual and text representations*
  $\mathbf{z}^{(I, \boldsymbol{P_v^*})} \leftarrow \mathcal{I}(\boldsymbol{e}(\mathbf{x}, \boldsymbol{P_v^*}); \boldsymbol{\theta}_{\mathcal{I}})$;
  $\mathbf{z}^{(t_i, \boldsymbol{P_t^*})} \leftarrow \mathcal{T}(\boldsymbol{w}(t_i, \boldsymbol{P_t^*}); \boldsymbol{\theta}_{\mathcal{T}})$;
  *# Generate adversarial examples*
  $\tilde{\mathbf{x}} = \arg\max_{\tilde{\mathbf{x}} \in \mathcal{B}_\epsilon(\mathbf{x})} \mathcal{L}_{\text{CE}}\left(\cos(\tilde{\mathbf{z}}^{(I, \boldsymbol{P_v^*})}, \mathbf{z}^{(t, \boldsymbol{P_t^*})}), y\right)$;
  *# Find the index of the highest similarity score*
  `nat_idx` $\leftarrow \arg\max\left(\cos\left(\mathbf{z}^{(I, \boldsymbol{P_v^*})}, \mathbf{z}^{(t, \boldsymbol{P_t^*})}\right)\right)$;
  `adv_idx` $\leftarrow \arg\max\left(\cos\left(\tilde{\mathbf{z}}^{(I, \boldsymbol{P_v^*})}, \mathbf{z}^{(t, \boldsymbol{P_t^*})}\right)\right)$;
  **if** `nat_idx` $== y$ **then**
    `nat_correct` $\leftarrow$ `nat_correct` $+ 1$;
  **end if**
  **if** `adv_idx` $== y$ **then**
    `adv_correct` $\leftarrow$ `adv_correct` $+ 1$;
  **end if**
**end for**
`nat_acc` $\leftarrow$ `nat_correct`$/n$;
`adv_acc` $\leftarrow$ `adv_correct`$/n$.

---

# B  Related Work

**Adversarial robustness.**  Adversarial attacks fool models by overlaying carefully designed imperceptible perturbations on input data [1, 2, 43]. In response to the susceptibility of models to such attacks, adversarial training [2, 44, 31, 45–49] has emerged as one of the most effective empirical defense methods to enhance model robustness. It incorporates adversarial data into the training process and ensures the model's predictive distribution for adversarial images closely aligns with the ground truth label. Moreover, recent advancements have seen the incorporation of contrastive learning into adversarial training [50–52], which enables models to learn robust feature representations through instance discrimination tasks. As a result, models can align predictions for natural and adversarial image pairs in a self-supervised manner [53–59]. Additionally, there's a growing interest in aligning predictions for adversarial image-text pairs in a text-supervised context [11, 60], offering new avenues for zero-shot adversarial evaluation. Nevertheless, current research utilizes CLIP text encoding to produce static text supervision, which, although effective for clean images, may not adequately cater to the nuances of adversarial examples.

**Adversarial few/zero-shot classification.** Adversarial training possesses a significantly larger sample complexity of robust generalization [61], making it challenging to learn robust representations from sparse data. Existing works in adversarial few-shot classification fall into two categories: meta-learning based [62, 63, 21], which optimize an adversarial meta-learner using both clean and adversarial examples, and non-meta-learning based [22, 64], employing strategies like auxiliary corrective classifiers [22, 64] or reweighted mechanisms [22] for learning robust embeddings. Additionally, Yucel et al. [65] initiated the investigation of adversarial robustness in a zero-shot learning setting, where no downstream statistics are available during training. Inspired by the successes of Vision Language Models (VLMs), recent studies [11, 66] have unanimously chosen to incorporate semantic information from text supervision to bridge the generalization gap.

**Vision-language models (VLMs).**  Foundational VLMs [12–19, 67, 68] integrate interactions derived from image and text encodings for multi-modal pre-training. Depending on their specific objectives, VLMs can be trained through image-text contrastive learning [12, 13, 15–17, 69, 19], image-text matching [17, 19], and text generation [17–19]. Utilizing large-scale image-text datasets (e.g., 400M pairs for CLIP [12], 1B for ALIGN [13]) and end-to-end pre-training, these models acquire semantic relations between text and image features, thus exhibiting a profound understanding of open-vocabulary concepts. Consequently, VLMs have emerged as state-of-the-art solutions for various visual and vision-language tasks [70–76]. Nevertheless, some recent researches [77, 78] reveal that VLMs are also highly susceptible to adversarial perturbations.

**Prompt learning for VLMs.**  Prompt learning, initially introduced in the NLP community [23–25], involves adapting pre-trained models by adding a small number of new learnable parameters in the input data for downstream tasks, without altering the pre-trained weights. This method stands out among other lightweight adaptation approaches due to its exceptional adaptability and flexibility [79]. It has garnered increasing attention for adapting vision [26, 27, 80] and vision-language models [28, 33, 81, 82, 30, 83, 84]. Specifically, in VLMs, CoOp [28] pioneers prompt engineering for adapting CLIP models by modeling learnable context vectors to replace hand-crafted text prompts. CoCoOp [33] further enhances the generalization ability of CoOp by introducing conditional prompts specific to each visual input instance. MaPLe [30] integrates vision and language prompts with inner synergy for cross-modality prompt learning. Two recent works, ProGrad [84] and PromptSRC [83], concurrently advance the generalization of prompt learning by employing regulating constraints from zero-shot CLIP predictions to prevent the forgetting of general knowledge.

# C  Additional Implementation Details

All experiments are conducted in an environment running PyTorch 1.10.1 and CUDA 11.3 on Python 3.8. Experiments of adversarial prompt tuning on the ImageNet-1K dataset are carried out on a single NVIDIA RTX A40 GPU, while experiments on the other 10 datasets are performed on a single NVIDIA RTX 4090 GPU.

## C.1 Additional Implementation Details for Baselines

**Adversarial visual prompt.** We implement the adversarial visual prompt following all architectural and parameter settings in [11] for a fair comparison. In detail, we follow their code implementation to use a token-level prompt with size 5 and an image padding prompt for 30 pixels around the image. An SGD optimizer and a consine learning rate scheduler are used to train 10 epochs with an initial learning rate of 40.

**Adversarial text prompt.** We adopt a CoOp architecture [28] as our text prompt baseline and adapt learnable context vectors with adversarial examples. We typically follow [28] to use 16 context tokens with an additional class token appended at the end of the context vector. An SGD optimizer and a consine learning rate scheduler are used to train 200 epochs with an initial learning rate of 0.002, which aligns with the training settings in CoOp.

**Adversarial multi-modal prompt.** Adversarial multi-modal prompt in this work follows all the design choices as MaPLe [30], but are adapted with an adversarial text-image contrastive loss. To sum up, it contains a token-level learnable token with size 2 in both text and visual branches in the first 9 transformer layers, and the deep prompts are coupled through a text-to-image projection. The above prompt tokens as well as the deep projections are optimized for 10 epochs with SGD optimizer and cosine learning rate scheduler from an initial learning rate of 0.0035.

**Adversarial vision language prompt.** Adversarial vision language prompts possess the same vision and language prompt design as adversarial multi-modal prompts, but vision and language prompts are independently adapted without interaction. All learnable prompts are adapted for 10 epochs with SGD optimizer and cosine learning rate scheduler from an initial learning rate of 0.0035.

**Overall methodological explanations.** We summarize the prompt design and loss function of both baselines and our methods in Table 6. Note that the prompt design for baselines follows the original settings in their corresponding paper, while we replace their loss function with the TeCoA loss for adversarial training and evaluation. This is consistent with the methods used in Mao et al. [11].

Table 6: Overall methodological explanations of baselines and our methods.

| Method | Visual Prompt Tokens | Text Prompt Tokens | Prompt Projections | Deep Prompts | Training Loss | Attack-time Loss |
|---|---|---|---|---|---|---|
| AdvVP | 5[1] | ✗ | ✗ | ✗ | TeCoA | TeCoA |
| AdvTP | ✗ | 16 | ✗ | ✗ | TeCoA | TeCoA |
| AdvVLP | 2 | 2 | ✗ | ✔ | TeCoA | TeCoA |
| AdvMaPLe | 2 | 2 | ✔ | ✔ | TeCoA | TeCoA |
| **FAP** | 2 | 2 | ✔ | ✔ | Eq.(7) | TeCoA |

## C.2 Hand-crafted Prompt Templates

We report the hand-crafted prompt templates used in Zero-shot CLIP, AdvVP, and our method for initialization on 11 image recognition datasets in Table 7.

Table 7: Hand-crafted text template for static text supervision of different datasets.

| Dataset | Template |
|---|---|
| ImageNet-1K | `"a photo of a {}."` |
| Caltech101 | `"a photo of a {}."` |
| DTD | `"{} texture."` |
| EuroSAT | `"a centered satellite photo of {}."` |
| OxfordPets | `"a photo of a {}, a type of pet."` |
| FGVCAircraft | `"a photo of a {}, a type of aircraft."` |
| Food101 | `"a photo of a {}, a type of food."` |
| Flowers102 | `"a photo of a {}, a type of flower."` |
| StanfordCars | `"a photo of a {}."` |
| SUN397 | `"a photo of a {}."` |
| UCF101 | `"a photo of a person doing {}."` |

---

[1]With additional pixel-level pad prompt.

Table 8: Cross-dataset generalization from ImageNet-1K to various downstream recognition datasets. We report the mean and standard deviation of natural and robust (PGD-100) accuracy. Bolded numbers denote the state-of-the-art results.

| Natural Acc (%) | ImageNet-1K | Caltech101 | DTD | EuroSAT | OxfordPets | FGVCAircraft | Food101 | Flowers102 | StanfordCars | SUN397 | UCF101 | Average |
|---|---|---|---|---|---|---|---|---|---|---|---|---|
| Zero-shot CLIP | **62.10** | **91.50** | **43.70** | **45.20** | **87.40** | **19.20** | **80.50** | **66.90** | **60.40** | **62.10** | **62.00** | **61.91** |
| AdvVp | 44.87±1.93 | 85.47±0.66 | 30.23±0.46 | 25.17±7.07 | 74.20±2.50 | 7.13±0.74 | 56.53±2.58 | 43.17±4.19 | 27.27±3.70 | 41.97±1.68 | 44.60±2.59 | 43.69±2.55 |
| AdvVLP | 53.23±0.58 | 87.33±0.31 | 33.43±0.73 | 18.37±0.29 | 78.80±0.82 | 10.70±0.59 | 55.80±1.56 | 49.77±0.73 | 38.70±0.45 | 52.80±0.57 | 51.50±0.65 | 48.22±0.66 |
| AdvMaPLe | 52.93±0.62 | 88.23±0.31 | 30.87±0.54 | 17.60±2.33 | 77.87±1.03 | 11.10±0.65 | 56.67±0.83 | 52.90±0.29 | 36.70±1.36 | 52.53±0.78 | 50.97±1.10 | 48.03±0.89 |
| **FAP** | 52.53±0.37 | 87.80±1.00 | 30.93±1.34 | 15.30±0.14 | 78.20±0.14 | 10.70±0.71 | 55.83±2.12 | 51.20±0.96 | 38.70±1.15 | 52.47±0.62 | 51.73±0.46 | 47.76±0.82 |
| | | | | | | | | | | | | |
| PGD-100 Acc (%) | ImageNet-1K | Caltech101 | DTD | EuroSAT | OxfordPets | FGVCAircraft | Food101 | Flowers102 | StanfordCars | SUN397 | UCF101 | Average |
| Zero-shot CLIP | 1.57±0.00 | 26.23±0.04 | 5.07±0.09 | 0.03±0.03 | 3.27±0.02 | 0.00±0.00 | 5.03±0.00 | 1.73±0.00 | 0.30±0.00 | 1.20±0.00 | 2.47±0.00 | 4.26±0.03 |
| AdvVp | 11.67±0.95 | 48.07±0.90 | 12.93±0.54 | 4.57±1.33 | 19.03±2.41 | 0.83±0.34 | 9.70±0.45 | 16.20±2.97 | 2.90±0.57 | 12.77±0.50 | 10.47±1.10 | 13.56±1.10 |
| AdvVLP | 22.10±0.36 | 62.97±0.74 | **18.60±0.24** | **10.67±0.45** | 40.83±2.02 | 2.73±0.46 | 17.83±0.90 | 25.23±1.22 | 10.97±0.26 | 21.67±0.39 | 22.10±0.96 | 23.25±0.73 |
| AdvMaPLe | 21.90±0.50 | 64.90±1.10 | 17.50±0.22 | 10.53±0.68 | 42.83±2.13 | 2.73±0.24 | 18.53±0.66 | **28.73±0.79** | 10.43±0.12 | 21.90±0.36 | 23.20±0.78 | 23.93±0.69 |
| **FAP** | **22.90±0.85** | **65.43±1.76** | 16.93±0.97 | 9.97±1.05 | **43.77±1.32** | **2.77±0.33** | **19.60±1.34** | 27.23±1.06 | **11.80±0.91** | **22.40±1.08** | **23.77±0.90** | **24.23±1.05** |

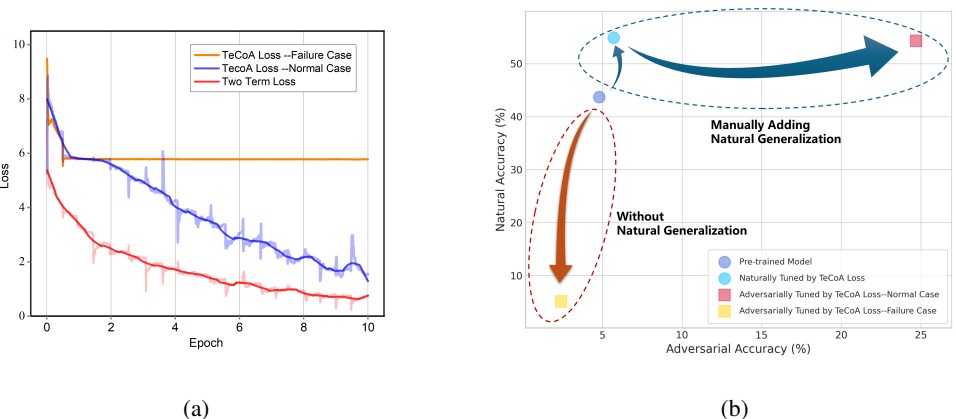

(a)  (b)

Figure 6: Illustration of potential failure cases and their solutions. Experiments of failure cases originate from 8-shot adversarial prompt learning on the DTD dataset.

# D    Additional Experimental Results

## D.1    Detailed Results on Adversarial Cross-Dataset Evaluation

For the cross-dataset evaluation, models are adapted on the ImageNet-1K dataset using 16 shots and then assessed for their zero-shot adversarial robustness across 10 distinct datasets, without further downstream tuning. As shown in Table 8, our method outperforms its counterparts in 8/11 datasets and baseline in all 11 datasets. Moreover, it reveals that robust adaptation takes the cost of natural accuracy, as models obtained using various robust adaptation methods exhibit a decline in zero-shot natural accuracy performance on downstream datasets, compared to the original CLIP model.

## D.2    Natural Generalization Gap Hinders Robust Adapting

We identify a failure risk in few-shot adversarial prompt learning using TeCoA loss [11], where insufficient natural generalization on the sampled dataset impedes robustness learning. Figure 6a displays the loss variation during training under this setup. Under the same experimental setup using the TeCoA loss, different trials exhibit completely different trends: the curve for the failure case shows that the loss quickly ceases to decline and becomes stable shortly after training begins, whereas the loss in the normal case continues to decrease as the training progresses.

We presume that this failure stems from a lack of natural generalization ability. To confirm this, we first conduct natural tuning on the problematic few-shot dataset and then apply adversarial prompt learning. This restores the model's robust fine-tuning performance, as evident in Figure 6b, where natural and robust accuracies improve significantly after natural example adaptation. Besides, we validate the learning process on the same few-shot dataset with a dual-form loss in the training objective that considers both natural and adversarial terms (red lines in Figure 6a). It is revealed that

this two-term loss effectively acts as a surrogate for the aforementioned two-stage method, avoiding potential failures caused by the natural generalization barrier in end-to-end training.

## D.3 Incremental Changes from AdvMaPLe

By examining the structural vulnerabilities (Figure 4) and the inadequate natural generalization (Figure 6) inherent in AdvMaPLe's learning objectives, we have proposed straightforward yet effective improvements. We generally have two improvements from AdvMaPLe:

- **Imp.1:** Regarding the prompt design, we optimize the projection direction under the adversarial prompt learning situation for superior stability.

- **Imp.2:** Regarding the learning objective, we not only consider the natural generalization gap that may cause the adversarial prompt to fail in the few-shot setting, but also make full use of the CLIP structure to design a differentiated robust learning strategy between different modalities.

For a clear picture of the empirical boost, we demonstrate the incremental changes concerning AdvMaPLe. We separately report the performance changes resulting from modifications to prompt direction alone, the learning objective alone, and the combination of both in Table 9.

We find from Table 9 that adopting our provided learning objective alone can enhance model performance. However, the performance change brought about by modifying the projection direction alone on the basis of AdvMaPLe is subtle, as the model's performance on most downstream datasets is not saturated at this point. On the other hand, further modifications to the projection direction based on row 3 can lead to additional improvements in model performance due to the repair of instabilities on certain datasets.

Table 9: Incremental changes with respect to AdvMaPLe. Our method combines Imp.1 and Imp.2 based on AdvMaPLe, achieving a significant performance improvement (results in the last row).

| Method | Prompt Direction | Training Objective | Base Nat Acc | Base Adv Acc | New Nat Acc | New Adv Acc |
|---|---|---|---|---|---|---|
| AdvMaPLe | $P_t \rightarrow P_v$ | TeCoA | 58.01 | 30.66 | 43.06 | 18.68 |
| +Imp.1 | $P_v \rightarrow P_t$ | TeCoA | 57.96 | 30.10 | 43.73 | 19.01 |
| +Imp.2 | $P_t \rightarrow P_v$ | Eq. (7) | 64.81 | 35.26 | 45.01 | 20.25 |
| +Imp.1&Imp.2 | $P_v \rightarrow P_t$ | Eq. (7) | **70.60** | **39.15** | **51.79** | **23.65** |

## D.4 Case Study: Improving AdvVLP with Our Learning Objective

We further illustrate the adversarial robustness enhancement brought by using our proposed training objective for prompt learning through an intuitive case study. Here, we adapt AdvVLP with both TeCoA loss and our $\mathcal{L}_{\text{final}}$. In Figure 7, our loss improves zero-shot adversarial robustness across ten out of eleven datasets.

Additionally, our training objective results in evident performance gain under few-shot base-to-new generalization, as revealed in Table 10. That is, we not only achieve better base natural accuracy (+11.11%), base PGD-100 accuracy (+6.67%), new natural accuracy (+2.03%), new PGD-100 accuracy (+0.87%), but also maintains superior stability across different trails.

## D.5 Results on Different CLIP Architectures

We provide the adversarial cross-dataset transfer results on another CLIP ViT backbone, ViT-B/16, that is adapted to the proposed method. With the same architectural design, ViT-B/16 divides the input image into smaller patches to better capture and learn image details. This makes ViT-B/16 generally have superior performance over ViT-B/32 in natural image recognition due to its finer granularity, but it also incurs higher computational costs due to longer input sequences. However, when considering tasks involving adversarial robustness, more complex models do not necessarily yield better performance [85]. We report the results on ViT-B/16 in Table 11. We find that ViT-B/16 does not bring about improved robustness performance, which is due to adversarial prompt learning focusing more on feature alignment and understanding between different modalities rather than

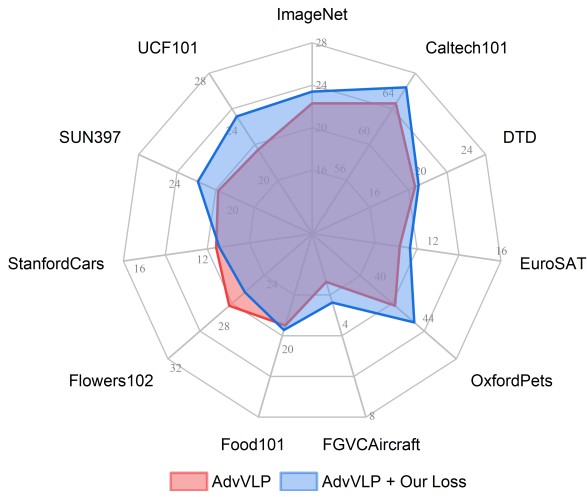

Figure 7: Zero-shot adversarial robustness of AdvVLP adapted with TeCoA loss (red) and our loss (blue).

Table 10: Few-shot base-to-new transfer results (%) on AdvVLP with different learning objectives. We also report the performance gains achieved by adapting with our $\mathcal{L}_{\text{final}}$.

| Metric | AdvVLP | AdvVLP + $\mathcal{L}_{\text{final}}$ |
|---|---|---|
| Base Nat Acc | 58.95±11.67 | 70.06±1.30 |
| $\Delta$ | | + 11.11 |
| Base Adv Acc | 32.37±6.67 | 39.04±1.42 |
| $\Delta$ | | + 6.67 |
| New Nat Acc | 46.92±7.41 | 48.95±2.17 |
| $\Delta$ | | + 2.03 |
| New Adv Acc | 21.61±3.86 | 22.48±1.96 |
| $\Delta$ | | + 0.87 |

detailed features. Therefore, the loss of detailed information resulting from the division of patches in ViT-B/32 is acceptable.

Table 11: Cross dataset transfer results on ViT-B/16. We report the natural and zero-shot PGD-100 accuracy (%) on the source ImageNet-1K dataset and 10 downstream target datasets.

| | Source | Target | | | | | | | | | | |
|---|---|---|---|---|---|---|---|---|---|---|---|---|
| ViT-B/16 | ImageNet-1K | Caltech101 | DTD | EuroSAT | OxfordPets | FGVCAircraft | Food101 | Flowers102 | StanfordCars | SUN397 | UCF101 | Average |
| Natural Accuracy | 55.40 | 86.90 | 25.00 | 15.00 | 77.40 | 12.50 | 51.90 | 45.80 | 38.50 | 50.00 | 48.70 | 45.84 |
| PGD-100 Accuracy | 24.50 | 63.70 | 13.20 | 10.70 | 45.80 | 4.70 | 16.20 | 22.30 | 10.80 | 20.50 | 19.50 | 23.24 |

## D.6 Zero-shot Adversarial Robustness under Different Perturbation Bounds

In this task, we provide adversarial attacks of varying intensities by changing the perturbation bounds to test the effectiveness of the model in learning robust representations from different adversarial distributions. Specifically, we set $\epsilon = \{1/255, 2/255, 4/255\}$ during the training phase respectively, and use the same $\epsilon$ values during testing as were used in training.

As can be seen in Figure 8, a larger perturbation bound brings a stronger attack, thus decreasing the zero-shot robust performance. As a lightweight adaptation method, prompt tuning for superior zero-shot adversarial robustness to large attack strength requires more training data.

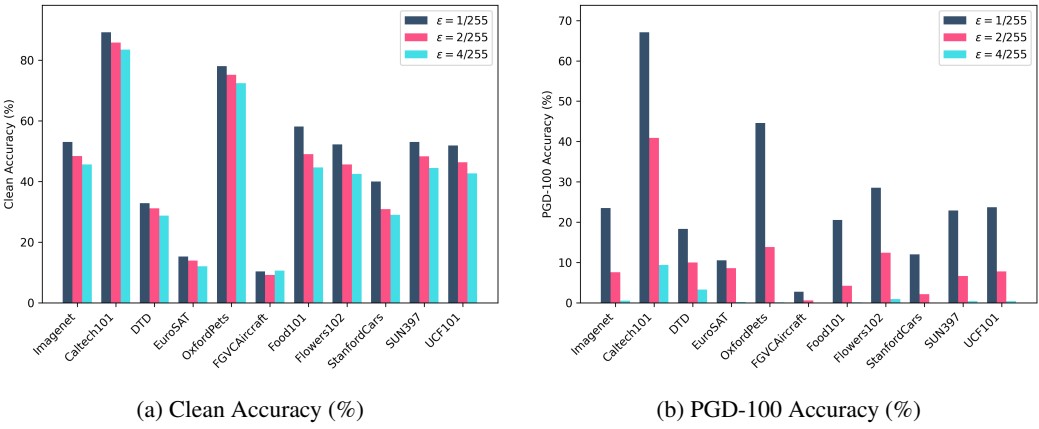

(a) Clean Accuracy (%)  (b) PGD-100 Accuracy (%)

Figure 8: Zero-shot adversarial robustness under different perturbation bounds.

### D.7 Zero-shot Adversarial Evaluation under Auto-Attack

We consider more powerful Auto-Attack [86] to evaluate our adapted model. Now that adversarial prompt tuning does not rely on the obfuscated gradient, we use two APGD variants, APGD-CE and APGD-DLR, in Auto-Attack to evaluate our models. In Table 12, we can conclude that Auto-Attack provides a stronger attack and causes varying degrees of performance degradation in each model. Our model still exhibits better robustness to Auto-Attack compared with AdvVP, AdvVLP, and AdvMaPLe. Moreover, by adapting AdvVLP with our learning objective in Appendix D.4, we achieve further performance gain under all three different perturbation bound settings. Note that Auto-Attack uses a fractional attack generator which explores that fraction space by automatically adjusting step size $\alpha$, it serves as a more effective and powerful attacker for zero-shot adversarial robustness evaluation.

Table 12: Zero-shot adversarial robustness (%) on downstream datasets with Auto-Attack adversarial perturbation. We consider different perturbation bounds $\epsilon = 1/255, 2/255, 4/255$ to evaluate models with different attack strengths. The best accuracies are bolded.

| $\epsilon = 1/255$ | ImageNet-1K | Caltech101 | DTD | EuroSAT | OxfordPets | FGVCAircraft | Food101 | Flowers102 | StanfordCars | SUN397 | UCF101 | Average |
|---|---|---|---|---|---|---|---|---|---|---|---|---|
| AdvVP | 10.64 | 47.27 | 8.62 | 1.88 | 17.32 | 1.06 | 6.98 | 15.62 | **6.64** | 12.38 | 9.29 | 12.52 |
| AdvMaPLe | 13.01 | 60.60 | 13.34 | 3.72 | 26.24 | 2.71 | **8.92** | 21.97 | 6.64 | **16.79** | **17.42** | 17.40 |
| AdvVLP | 12.99 | 60.25 | 13.62 | 4.90 | 26.40 | **2.97** | 7.71 | 20.01 | 5.31 | 16.54 | 16.07 | 16.98 |
| **FAP** | **13.95** | **61.17** | 14.29 | 1.17 | 30.19 | 2.40 | 8.83 | **22.52** | 4.95 | 15.66 | 16.41 | 17.41 |
| **FAP (+AdvVLP)** | 12.93 | 59.01 | 15.94 | **12.70** | **30.40** | 2.32 | 8.13 | 18.11 | 4.50 | 15.90 | 15.57 | **17.77** |

| $\epsilon = 2/255$ | ImageNet-1K | Caltech101 | DTD | EuroSAT | OxfordPets | FGVCAircraft | Food101 | Flowers102 | StanfordCars | SUN397 | UCF101 | Average |
|---|---|---|---|---|---|---|---|---|---|---|---|---|
| AdvVP | 4.23 | 29.83 | 5.62 | 1.35 | 3.98 | 0.24 | 1.55 | 5.34 | 1.39 | 3.88 | 2.67 | 5.46 |
| AdvMaPLe | 10.41 | 55.90 | 11.89 | 1.74 | 19.00 | 1.90 | 6.31 | 18.49 | **4.62** | 12.99 | **13.71** | 14.27 |
| AdvVLP | 10.30 | 55.16 | 12.50 | 1.96 | 19.13 | **2.26** | 5.61 | 17.84 | 3.47 | **13.02** | 12.15 | 13.95 |
| **FAP** | **11.13** | **56.78** | 12.87 | 0.44 | 22.82 | 1.93 | **6.32** | **18.65** | 3.51 | 12.29 | 12.22 | 14.45 |
| **FAP(+AdvVLP)** | 10.21 | 54.78 | **14.28** | **11.23** | **23.11** | 1.91 | 5.91 | 16.11 | 3.41 | 12.52 | 12.31 | **15.07** |

| $\epsilon = 4/255$ | ImageNet-1K | Caltech101 | DTD | EuroSAT | OxfordPets | FGVCAircraft | Food101 | Flowers102 | StanfordCars | SUN397 | UCF101 | Average |
|---|---|---|---|---|---|---|---|---|---|---|---|---|
| AdvVP | 1.71 | 15.28 | 2.07 | 0.70 | 0.34 | 0.12 | 0.11 | 0.46 | 0.13 | 0.62 | 0.34 | 1.99 |
| AdvMaPLe | 6.32 | 46.12 | 9.47 | 0.32 | 9.48 | 1.03 | **3.28** | 12.35 | **2.24** | 7.31 | 7.24 | 9.56 |
| AdvVLP | 6.35 | 46.30 | 10.02 | 0.30 | 9.26 | **1.32** | 3.00 | 11.93 | 1.41 | **7.58** | 6.22 | 9.43 |
| **FAP** | **7.01** | **48.27** | 10.47 | 0.11 | 12.27 | 1.09 | 3.21 | **13.47** | 1.76 | 7.45 | **7.38** | 10.23 |
| **FAP (+AdvVLP)** | 6.30 | 46.08 | **11.45** | **9.07** | **13.25** | 1.00 | 3.16 | 11.85 | 1.68 | 7.50 | 6.48 | **10.71** |

### D.8 Discussions on Training-time Attack Generation

We adopt Eq. (8) to carry out adversarial attacks during the training process. We did not take $\mathcal{L}_{\cos}$ into account in Eq. (8). Including $\mathcal{L}_{\cos}$ in the generation of adversarial samples would make the gradient information focus on the differences between natural and adversarial examples, thereby generating stronger adversarial perturbations with greater differences from the natural examples. However, since we have incorporated this term in our adversarial defense, the model will gradually provide stronger attacks during iterative learning to ensure differences in image features between natural and adversarial samples, making it somewhat redundant in function.

We validate this with the experimental results in Table 13. We can observe that the results of these two methods for generating adversarial attacks are quite similar, indicating that adding $\mathcal{L}_{\cos}$ in the attack is indeed redundant. Therefore, for the sake of simplicity, we did not include $\mathcal{L}_{\cos}$ for training-time attack generation.

Table 13: Comparison in train-time attack generation methods.

| Train-time attack generation | Base Nat Acc | Base Adv Acc | New Nat Acc | New Adv Acc |
|:---:|:---:|:---:|:---:|:---:|
| $\mathcal{L}_{\mathrm{KL}}$ | **70.52±0.82** | 38.05±2.15 | **49.58±3.55** | 21.86±2.57 |
| $\mathcal{L}_{\cos} \cdot \mathcal{L}_{\mathrm{KL}}$ | 70.04±0.94 | **38.06±2.23** | 49.56±3.00 | **22.03±2.19** |

### D.9 Detailed Results for Adversarial Few-shot Learning

For adversarial few-shot prompt learning, we plot curves showing how the average natural and robust accuracy change with varying shot numbers in Figure 3. Here, we present the mean and standard deviation of natural (Table 14) and robust (Table 15) accuracy for all experimental settings, datasets, and shot numbers, based on our multiple trials. For our proposed method, when given a smaller number of training samples, both the standard deviation of natural accuracy and robust accuracy are relatively high, indicating that the performance of learning robust representations at this stage depends on the quality of the examples. As the shot number increases, our method exhibits a significant reduction in the standard deviation for both natural and robust accuracy, demonstrating its ability to acquire adversarial robustness stability.

### D.10 Detailed Results for Adversarial Base-to-New Generalization

For adversarial base-to-new generalization results in Section 4.2, we further provide the detailed results on each dataset. In Table 16, our method demonstrates preferable learning performance on the majority of datasets. Specifically, in recognition datasets for fine-grained tasks that significantly differ from generic knowledge (DTD, Flowers102, OxfordPets, FGVCAircraft, etc.), our training objective effectively avoids potential failures caused by natural generalization barriers in robustness learning, thus yielding more stable results across multiple trials.

### D.11 Comparison between Adversarial Text and Vision Prompt

We design most of the baseline settings on the top of the adversarial vision prompt framework. As a result, most of them belong to a cross-modal prompt family, with learnable prompt tokens not only exist in both vision and text input sequences. However, for completeness, we also consider the design of prompts in a uni-modal context, namely adversarial vision prompts (AdvVP) and adversarial text prompts (AdvTP). In Figure 9, we find that, as the number of available examples increases, both vision and text prompts fail to acquire more robustness correlated hints for promoting adversarial robustness. However, although it seems difficult for AdvTP to learn proper adversarial text supervision, AdvTP is capable of maintaining preferable natural performance even when only adversarial examples are visible. We believe this can be attributed to the text prompt's ability to capture semantic information.

## E   Impact Statement

This research aims to contribute positively to the machine learning field by enhancing model robustness against adversarial attacks. While we believe our work is unlikely to have direct negative societal impacts, we acknowledge the importance of considering potential misuse scenarios, such as in the context of security applications. The broader implication of our study is that it enables neural models to maintain adversarial robustness with minimal adaptations, making it particularly suitable for real-time applications in mobile and embodied systems. Such advancements could lead to more secure and reliable applications in various real-world scenarios, including mobile device security.

Table 14: Natural Accuracy (%) of detailed adversarial few-shot prompt learning results. We report the mean and standard deviation of the natural accuracy for baselines and our method under different shot number settings across 11 datasets.

| Dataset | Method | 1-shot | 2-shot | 4-shot | 8-shot | 16-shot |
|---|---|---|---|---|---|---|
| Average | AdvVP | 32.81 ± 3.37 | 32.87 ± 5.99 | 34.13 ± 8.24 | 34.00 ± 6.02 | 33.59 ± 4.71 |
| | AdvTP | 52.02 ± 1.55 | 52.85 ± 3.20 | 56.42 ± 1.11 | 58.68 ± 0.41 | 60.73 ± 0.51 |
| | AdvMaPLe | 28.22 ± 4.99 | 34.18 ± 1.69 | 44.05 ± 5.22 | 54.65 ± 2.85 | 64.24 ± 1.28 |
| | AdvVLP | 28.47 ± 1.73 | 37.22 ± 0.80 | 46.70 ± 4.23 | 56.64 ± 1.16 | 58.62 ± 2.19 |
| | FAP | 35.42 ± 7.44 | 48.17 ± 1.86 | 53.38 ± 3.33 | 62.17 ± 0.34 | 65.32 ± 0.08 |
| ImageNet-1K | AdvVP | 46.60 ± 3.77 | 46.93 ± 2.21 | 49.80 ± 1.69 | 46.37 ± 0.62 | 46.27 ± 1.46 |
| | AdvTP | 49.30 ± 1.34 | 48.83 ± 0.68 | 50.90 ± 0.37 | 52.03 ± 0.50 | 52.63 ± 0.37 |
| | AdvMaPLe | 49.27 ± 0.45 | 49.97 ± 0.54 | 51.27 ± 0.83 | 52.13 ± 0.58 | 52.93 ± 0.62 |
| | AdvVLP | 49.00 ± 1.13 | 50.53 ± 1.08 | 51.30 ± 0.71 | 52.83 ± 0.12 | 53.23 ± 0.58 |
| | FAP | 49.90 ± 0.51 | 48.53 ± 0.90 | 51.53 ± 1.21 | 52.17 ± 0.45 | 52.53 ± 0.37 |
| Caltech101 | AdvVP | 85.73 ± 7.00 | 91.23 ± 0.21 | 90.17 ± 0.87 | 90.30 ± 0.29 | 90.40 ± 0.42 |
| | AdvTP | 84.77 ± 5.56 | 89.70 ± 0.43 | 90.77 ± 0.70 | 92.37 ± 0.53 | 92.93 ± 0.29 |
| | AdvMaPLe | 85.53 ± 1.35 | 88.00 ± 0.71 | 89.53 ± 0.65 | 90.63 ± 0.37 | 92.17 ± 0.21 |
| | AdvVLP | 85.43 ± 2.21 | 87.60 ± 0.65 | 89.37 ± 0.70 | 90.17 ± 0.90 | 92.37 ± 0.12 |
| | FAP | 83.53 ± 4.06 | 87.73 ± 0.49 | 87.57 ± 0.09 | 89.63 ± 0.95 | 91.10 ± 0.42 |
| DTD | AdvVP | 26.97 ± 11.64 | 14.27 ± 2.52 | 18.77 ± 0.09 | 23.63 ± 0.71 | 29.20 ± 9.73 |
| | AdvTP | 41.67 ± 1.27 | 45.57 ± 1.39 | 51.33 ± 1.17 | 54.43 ± 1.11 | 54.50 ± 0.43 |
| | AdvMaPLe | 13.63 ± 13.66 | 16.53 ± 16.42 | 6.43 ± 0.95 | 33.20 ± 19.91 | 57.93 ± 0.78 |
| | AdvVLP | 15.97 ± 15.33 | 18.33 ± 15.90 | 22.97 ± 13.33 | 51.83 ± 1.16 | 57.53 ± 0.66 |
| | FAP | 18.40 ± 11.94 | 18.40 ± 16.59 | 31.27 ± 17.60 | 52.13 ± 0.68 | 55.17 ± 1.14 |
| EuroSAT | AdvVP | 9.87 ± 0.87 | 9.83 ± 0.50 | 10.57 ± 0.56 | 9.87 ± 0.87 | 18.13 ± 5.96 |
| | AdvTP | 40.47 ± 12.54 | 40.87 ± 10.90 | 25.67 ± 11.51 | 24.33 ± 4.00 | 33.40 ± 3.94 |
| | AdvMaPLe | 15.10 ± 2.81 | 21.57 ± 6.45 | 29.27 ± 5.82 | 27.07 ± 5.62 | 54.97 ± 12.19 |
| | AdvVLP | 14.37 ± 2.39 | 20.37 ± 4.32 | 13.20 ± 3.78 | 10.87 ± 0.71 | 15.50 ± 3.96 |
| | FAP | 31.37 ± 7.97 | 43.80 ± 15.10 | 64.37 ± 9.85 | 76.57 ± 3.92 | 81.70 ± 1.10 |
| OxfordPets | AdvVP | 57.60 ± 38.19 | 47.13 ± 33.94 | 57.80 ± 38.19 | 57.43 ± 38.07 | 56.40 ± 38.18 |
| | AdvTP | 70.23 ± 2.60 | 72.87 ± 1.33 | 71.83 ± 9.43 | 82.87 ± 0.46 | 83.70 ± 0.99 |
| | AdvMaPLe | 30.67 ± 34.32 | 34.03 ± 31.37 | 30.70 ± 35.81 | 55.60 ± 36.70 | 83.27 ± 0.57 |
| | AdvVLP | 29.63 ± 31.17 | 31.27 ± 29.44 | 67.43 ± 9.83 | 80.67 ± 0.54 | 82.93 ± 0.29 |
| | FAP | 49.23 ± 25.72 | 64.23 ± 19.91 | 42.10 ± 29.52 | 79.47 ± 0.45 | 81.90 ± 0.85 |
| FGVCAircraft | AdvVP | 1.50 ± 0.36 | 5.97 ± 6.47 | 6.10 ± 7.00 | 4.70 ± 5.23 | 1.33 ± 0.24 |
| | AdvTP | 14.77 ± 1.68 | 16.37 ± 1.43 | 15.70 ± 1.07 | 13.60 ± 1.27 | 14.77 ± 0.74 |
| | AdvMaPLe | 1.37 ± 0.12 | 1.80 ± 0.50 | 2.50 ± 0.45 | 20.37 ± 1.44 | 23.63 ± 0.98 |
| | AdvVLP | 1.90 ± 0.70 | 6.70 ± 4.68 | 14.07 ± 4.12 | 14.70 ± 9.82 | 23.27 ± 0.88 |
| | FAP | 2.37 ± 0.39 | 9.57 ± 4.91 | 19.57 ± 0.21 | 21.03 ± 0.34 | 23.50 ± 0.36 |
| Food101 | AdvVP | 24.43 ± 32.64 | 1.03 ± 0.05 | 22.73 ± 30.66 | 1.00 ± 0.00 | 1.07 ± 0.09 |
| | AdvTP | 56.57 ± 1.94 | 60.17 ± 1.08 | 59.80 ± 1.30 | 61.57 ± 1.19 | 62.50 ± 1.85 |
| | AdvMaPLe | 5.27 ± 3.37 | 3.10 ± 0.88 | 60.00 ± 0.29 | 62.70 ± 0.29 | 65.13 ± 0.52 |
| | AdvVLP | 1.07 ± 0.09 | 1.53 ± 0.58 | 41.50 ± 25.81 | 61.73 ± 0.57 | 43.30 ± 29.85 |
| | FAP | 31.67 ± 22.98 | 56.90 ± 1.18 | 59.37 ± 0.74 | 61.80 ± 0.08 | 64.03 ± 0.69 |
| Flowers102 | AdvVP | 63.10 ± 1.22 | 61.47 ± 1.28 | 55.97 ± 0.74 | 55.50 ± 1.02 | 56.17 ± 0.61 |
| | AdvTP | 61.97 ± 4.65 | 67.17 ± 12.16 | 82.40 ± 0.57 | 84.00 ± 2.09 | 86.63 ± 0.33 |
| | AdvMaPLe | 1.40 ± 0.71 | 46.17 ± 29.83 | 52.20 ± 35.43 | 83.10 ± 0.62 | 87.87 ± 0.12 |
| | AdvVLP | 19.77 ± 26.40 | 62.43 ± 7.09 | 51.00 ± 35.57 | 83.90 ± 1.02 | 87.70 ± 0.51 |
| | FAP | 10.40 ± 2.35 | 53.10 ± 15.70 | 73.13 ± 0.58 | 81.53 ± 0.45 | 86.27 ± 0.66 |
| StanfordCars | AdvVP | 0.57 ± 0.0 | 31.20 ± 21.57 | 14.00 ± 18.88 | 14.40 ± 19.59 | 14.83 ± 20.34 |
| | AdvTP | 40.40 ± 1.42 | 15.57 ± 19.89 | 43.37 ± 1.31 | 49.43 ± 1.11 | 51.90 ± 0.67 |
| | AdvMaPLe | 25.80 ± 13.46 | 39.93 ± 0.81 | 44.60 ± 1.08 | 50.53 ± 0.31 | 56.17 ± 0.49 |
| | AdvVLP | 35.33 ± 0.54 | 40.07 ± 0.17 | 45.00 ± 0.65 | 50.93 ± 0.38 | 56.00 ± 1.00 |
| | FAP | 34.70 ± 1.24 | 38.60 ± 0.29 | 43.20 ± 0.45 | 48.47 ± 0.62 | 54.23 ± 0.61 |
| SUN397 | AdvVP | 41.20 ± 9.66 | 50.77 ± 7.06 | 48.47 ± 9.38 | 52.53 ± 0.81 | 54.70 ± 0.64 |
| | AdvTP | 53.53 ± 0.69 | 59.20 ± 0.16 | 62.37 ± 0.19 | 64.30 ± 0.43 | 65.67 ± 0.45 |
| | AdvMaPLe | 49.70 ± 0.29 | 53.73 ± 1.46 | 58.23 ± 0.05 | 61.50 ± 0.14 | 63.57 ± 0.31 |
| | AdvVLP | 48.83 ± 0.46 | 53.77 ± 1.25 | 57.90 ± 0.16 | 61.33 ± 0.39 | 63.90 ± 0.08 |
| | FAP | 49.53 ± 0.31 | 54.07 ± 0.33 | 56.60 ± 0.79 | 60.40 ± 0.62 | 62.37 ± 0.12 |
| UCF101 | AdvVP | 3.37 ± 2.79 | 1.73 ± 0.50 | 1.07 ± 0.33 | 18.27 ± 23.57 | 0.97 ± 0.21 |
| | AdvTP | 58.50 ± 0.45 | 65.00 ± 0.28 | 66.53 ± 1.96 | 66.53 ± 1.23 | 69.40 ± 0.85 |
| | AdvMaPLe | 32.70 ± 21.85 | 21.17 ± 24.36 | 59.73 ± 0.70 | 64.33 ± 1.10 | 68.97 ± 1.17 |
| | AdvVLP | 11.83 ± 5.10 | 36.83 ± 25.06 | 59.97 ± 1.18 | 64.07 ± 0.90 | 69.10 ± 0.73 |
| | FAP | 28.50 ± 20.56 | 54.93 ± 1.43 | 58.50 ± 1.59 | 60.70 ± 1.08 | 65.70 ± 0.28 |

Table 15: Robust Accuracy (%) of detailed adversarial few-shot prompt learning results. We report the mean and standard deviation of the PGD-100 accuracy for baselines and our method under different shot number settings across 11 datasets.

| Dataset | Method | 1-shot | 2-shot | 4-shot | 8-shot | 16-shot |
|---|---|---|---|---|---|---|
| **Average** | **AdvVP** | 14.04 ± 0.85 | 13.20 ± 1.73 | 13.08 ± 1.95 | 13.77 ± 1.42 | 14.28 ± 1.25 |
| | **AdvTP** | 3.75 ± 0.35 | 4.33 ± 0.21 | 4.55 ± 0.23 | 5.71 ± 0.07 | 6.42 ± 0.18 |
| | **AdvMaPLe** | 8.58 ± 1.17 | 12.36 ± 0.60 | 18.07 ± 1.72 | 25.78 ± 0.81 | 32.98 ± 0.56 |
| | **AdvVLP** | 9.01 ± 0.50 | 14.18 ± 0.16 | 18.80 ± 1.95 | 26.62 ± 0.23 | 30.84 ± 0.88 |
| | **FAP** | 7.88 ± 1.56 | 14.05 ± 1.05 | 19.59 ± 1.09 | 29.51 ± 0.42 | 34.61 ± 0.28 |
| **ImageNet-1K** | **AdvVP** | 11.07 ± 1.15 | 10.90 ± 0.45 | 11.13 ± 0.76 | 11.90 ± 0.71 | 12.77 ± 1.46 |
| | **AdvTP** | 1.30 ± 0.08 | 1.03 ± 0.05 | 1.40 ± 0.16 | 1.80 ± 0.08 | 2.07 ± 0.12 |
| | **AdvMaPLe** | 14.60 ± 0.14 | 17.13 ± 0.42 | 19.00 ± 0.29 | 20.60 ± 0.43 | 21.90 ± 0.50 |
| | **AdvVLP** | 15.53 ± 0.58 | 17.50 ± 0.22 | 19.37 ± 0.26 | 20.97 ± 0.05 | 22.10 ± 0.36 |
| | **FAP** | 15.40 ± 0.45 | 17.83 ± 0.47 | 19.60 ± 0.08 | 21.53 ± 0.21 | 22.90 ± 0.85 |
| **Caltech101** | **AdvVP** | 50.33 ± 6.74 | 55.23 ± 0.97 | 52.50 ± 0.42 | 50.33 ± 1.95 | 52.60 ± 1.14 |
| | **AdvTP** | 26.90 ± 5.35 | 31.70 ± 1.49 | 26.67 ± 1.58 | 30.83 ± 1.30 | 30.23 ± 1.02 |
| | **AdvMaPLe** | 48.37 ± 2.58 | 56.20 ± 0.83 | 59.40 ± 0.75 | 63.80 ± 0.92 | 68.63 ± 0.46 |
| | **AdvVLP** | 48.47 ± 3.08 | 55.33 ± 0.17 | 59.07 ± 0.68 | 63.13 ± 0.17 | 67.97 ± 1.04 |
| | **FAP** | 41.13 ± 7.58 | 53.90 ± 0.99 | 57.33 ± 0.48 | 62.50 ± 0.92 | 67.33 ± 1.25 |
| **DTD** | **AdvVP** | 12.93 ± 7.62 | 6.93 ± 0.74 | 9.27 ± 0.40 | 11.47 ± 0.37 | 13.87 ± 4.00 |
| | **AdvTP** | 3.83 ± 0.37 | 4.27 ± 1.03 | 6.33 ± 0.59 | 8.70 ± 0.50 | 10.47 ± 0.42 |
| | **AdvMaPLe** | 2.93 ± 3.72 | 4.20 ± 4.68 | 2.40 ± 1.36 | 16.97 ± 8.60 | 32.17 ± 0.34 |
| | **AdvVLP** | 4.77 ± 5.47 | 7.17 ± 6.61 | 10.33 ± 6.43 | 25.77 ± 0.40 | 32.73 ± 0.82 |
| | **FAP** | 2.40 ± 2.65 | 4.33 ± 5.85 | 8.07 ± 5.71 | 25.77 ± 0.98 | 31.33 ± 1.89 |
| **EuroSAT** | **AdvVP** | 9.80 ± 0.92 | 8.67 ± 0.97 | 8.50 ± 3.33 | 9.77 ± 0.96 | 15.83 ± 4.65 |
| | **AdvTP** | 0.30 ± 0.24 | 0.17 ± 0.12 | 0.27 ± 0.17 | 0.17 ± 0.17 | 0.87 ± 0.52 |
| | **AdvMaPLe** | 0.57 ± 0.46 | 5.37 ± 3.79 | 16.13 ± 7.40 | 21.60 ± 0.85 | 32.97 ± 5.88 |
| | **AdvVLP** | 0.20 ± 0.28 | 6.30 ± 4.61 | 6.83 ± 3.03 | 12.23 ± 1.75 | 17.30 ± 4.39 |
| | **FAP** | 0.00 ± 0.00 | 1.00 ± 1.41 | 3.60 ± 2.86 | 29.30 ± 2.96 | 39.73 ± 3.29 |
| **OxfordPets** | **AdvVP** | 22.73 ± 15.87 | 15.10 ± 10.34 | 16.20 ± 11.33 | 17.33 ± 11.97 | 16.43 ± 11.55 |
| | **AdvTP** | 0.60 ± 0.16 | 1.07 ± 0.50 | 2.10 ± 0.71 | 3.10 ± 0.80 | 4.40 ± 0.16 |
| | **AdvMaPLe** | 4.97 ± 6.81 | 6.87 ± 8.80 | 9.03 ± 10.45 | 21.07 ± 12.46 | 36.87 ± 0.78 |
| | **AdvVLP** | 3.83 ± 4.01 | 7.07 ± 8.32 | 18.47 ± 4.29 | 29.63 ± 0.34 | 35.57 ± 0.96 |
| | **FAP** | 3.47 ± 3.94 | 12.67 ± 8.69 | 9.30 ± 12.30 | 34.57 ± 1.19 | 41.00 ± 0.62 |
| **FGVCAircraft** | **AdvVP** | 0.77 ± 0.33 | 1.60 ± 0.71 | 1.27 ± 1.08 | 1.20 ± 0.43 | 0.63 ± 0.39 |
| | **AdvTP** | 0.10 ± 0.08 | 0.13 ± 0.09 | 0.67 ± 0.09 | 1.03 ± 0.09 | 1.27 ± 0.05 |
| | **AdvMaPLe** | 0.07 ± 0.09 | 0.73 ± 0.29 | 1.07 ± 0.29 | 5.53 ± 0.65 | 7.33 ± 0.12 |
| | **AdvVLP** | 0.90 ± 0.36 | 2.27 ± 0.60 | 3.73 ± 0.90 | 4.40 ± 2.41 | 8.40 ± 0.22 |
| | **FAP** | 0.07 ± 0.09 | 1.10 ± 1.28 | 3.93 ± 0.31 | 6.07 ± 0.29 | 7.97 ± 0.53 |
| **Food101** | **AdvVP** | 5.23 ± 6.56 | 0.10 ± 0.00 | 4.57 ± 5.68 | 0.83 ± 0.17 | 0.80 ± 0.28 |
| | **AdvTP** | 0.83 ± 0.25 | 0.87 ± 0.17 | 1.63 ± 0.09 | 2.33 ± 0.12 | 2.63 ± 0.05 |
| | **AdvMaPLe** | 0.30 ± 0.42 | 0.67 ± 0.46 | 14.83 ± 0.66 | 20.13 ± 0.53 | 25.27 ± 0.21 |
| | **AdvVLP** | 0.77 ± 0.21 | 1.10 ± 0.36 | 11.20 ± 6.11 | 19.33 ± 0.34 | 16.50 ± 10.83 |
| | **FAP** | 1.43 ± 1.82 | 10.53 ± 5.54 | 18.37 ± 0.21 | 23.20 ± 0.51 | 26.67 ± 0.40 |
| **Flowers102** | **AdvVP** | 29.70 ± 1.64 | 26.93 ± 0.31 | 23.73 ± 2.04 | 23.57 ± 0.54 | 22.03 ± 0.45 |
| | **AdvTP** | 2.10 ± 0.79 | 3.10 ± 0.80 | 4.23 ± 0.41 | 6.00 ± 0.29 | 8.97 ± 0.59 |
| | **AdvMaPLe** | 0.10 ± 0.08 | 17.00 ± 11.41 | 25.37 ± 17.02 | 48.80 ± 0.65 | 58.70 ± 1.00 |
| | **AdvVLP** | 6.57 ± 8.65 | 25.17 ± 2.83 | 25.80 ± 17.75 | 50.90 ± 0.50 | 58.70 ± 0.57 |
| | **FAP** | 0.53 ± 0.50 | 19.57 ± 12.73 | 38.77 ± 0.95 | 52.63 ± 1.25 | 61.47 ± 0.66 |
| **StanfordCars** | **AdvVP** | 0.33 ± 0.17 | 5.07 ± 3.71 | 2.93 ± 3.73 | 2.80 ± 3.75 | 3.57 ± 4.69 |
| | **AdvTP** | 0.23 ± 0.05 | 0.13 ± 0.19 | 0.83 ± 0.09 | 1.17 ± 0.05 | 1.60 ± 0.16 |
| | **AdvMaPLe** | 2.77 ± 0.99 | 5.20 ± 0.75 | 8.70 ± 0.42 | 12.80 ± 1.04 | 17.57 ± 0.53 |
| | **AdvVLP** | 3.80 ± 0.22 | 5.33 ± 0.56 | 9.07 ± 0.37 | 13.27 ± 0.29 | 17.47 ± 1.03 |
| | **FAP** | 4.83 ± 0.45 | 7.27 ± 0.24 | 11.17 ± 0.52 | 15.10 ± 0.49 | 19.23 ± 1.14 |
| **SUN397** | **AdvVP** | 11.10 ± 4.48 | 13.57 ± 3.18 | 13.03 ± 2.92 | 17.30 ± 0.73 | 17.63 ± 0.69 |
| | **AdvTP** | 1.23 ± 0.05 | 2.03 ± 0.09 | 2.90 ± 0.08 | 3.40 ± 0.00 | 3.67 ± 0.09 |
| | **AdvMaPLe** | 12.67 ± 0.24 | 16.33 ± 1.08 | 21.53 ± 0.59 | 26.30 ± 0.24 | 29.70 ± 0.24 |
| | **AdvVLP** | 12.60 ± 0.28 | 17.33 ± 0.59 | 21.17 ± 0.24 | 26.23 ± 0.19 | 29.70 ± 0.22 |
| | **FAP** | 14.93 ± 0.21 | 19.30 ± 0.59 | 23.20 ± 1.00 | 27.23 ± 0.12 | 30.27 ± 0.19 |
| **UCF101** | **AdvVP** | 0.40 ± 0.08 | 1.07 ± 0.12 | 0.80 ± 0.41 | 4.93 ± 5.85 | 0.93 ± 0.21 |
| | **AdvTP** | 3.87 ± 0.50 | 3.10 ± 0.37 | 3.03 ± 0.17 | 4.30 ± 0.29 | 4.40 ± 0.14 |
| | **AdvMaPLe** | 7.07 ± 4.62 | 6.20 ± 7.57 | 21.30 ± 0.51 | 25.93 ± 0.61 | 31.67 ± 0.97 |
| | **AdvVLP** | 1.73 ± 1.11 | 11.43 ± 7.17 | 21.77 ± 0.49 | 26.97 ± 1.39 | 32.80 ± 0.24 |
| | **FAP** | 2.43 ± 3.16 | 7.03 ± 5.92 | 22.13 ± 0.95 | 26.67 ± 0.48 | 32.80 ± 1.07 |

Table 16: Detailed results for base-to-new generalization on 11 datasets. We report the Natural and PGD-100 Accuracy (%) on the base and new classes that adapted with 16-shot adversarial prompt learning.

| Dataset | Class | Metric | AdvVP | AdvMaPLe | AdvVLP | FAP |
|---|---|---|---|---|---|---|
| **Average** | **Base** | Natural Acc | 31.68 ± 6.57 | 60.38 ± 8.03 | 58.95 ± 11.67 | 70.52 ± 0.82 |
| | | Adv Acc | 14.43 ± 2.26 | 30.69 ± 4.71 | 32.37 ± 6.67 | 38.05 ± 2.15 |
| | **New** | Natural Acc | 30.39 ± 6.40 | 46.18 ± 6.39 | 46.92 ± 7.41 | 49.58 ± 3.55 |
| | | Adv Acc | 13.36 ± 2.80 | 20.25 ± 3.39 | 21.61 ± 3.86 | 21.86 ± 2.57 |
| **ImageNet-1K** | **Base** | Natural Acc | 49.87 ± 1.70 | 58.40 ± 0.57 | 58.47 ± 0.25 | 58.10 ± 0.14 |
| | | Adv Acc | 12.27 ± 0.34 | 25.33 ± 0.19 | 24.93 ± 0.21 | 25.83 ± 0.09 |
| | **New** | Natural Acc | 44.80 ± 2.41 | 48.83 ± 0.90 | 48.67 ± 0.12 | 47.83 ± 0.31 |
| | | Adv Acc | 12.27 ± 0.52 | 21.03 ± 0.21 | 20.50 ± 0.08 | 21.57 ± 0.31 |
| **Caltech101** | **Base** | Natural Acc | 92.83 ± 0.91 | 94.40 ± 0.65 | 94.87 ± 0.17 | 94.07 ± 0.77 |
| | | Adv Acc | 57.17 ± 1.23 | 73.90 ± 0.14 | 76.23 ± 1.08 | 74.20 ± 1.73 |
| | **New** | Natural Acc | 88.83 ± 0.38 | 83.27 ± 1.27 | 84.47 ± 0.85 | 76.53 ± 2.60 |
| | | Adv Acc | 49.13 ± 1.79 | 56.70 ± 1.16 | 57.67 ± 1.06 | 50.00 ± 1.00 |
| **DTD** | **Base** | Natural Acc | 23.27 ± 5.49 | 43.40 ± 25.05 | 48.63 ± 24.86 | 69.17 ± 0.56 |
| | | Adv Acc | 10.03 ± 2.17 | 21.50 ± 14.25 | 27.57 ± 12.89 | 41.63 ± 2.12 |
| | **New** | Natural Acc | 13.23 ± 1.40 | 21.27 ± 12.11 | 22.87 ± 12.71 | 35.17 ± 7.71 |
| | | Adv Acc | 7.20 ± 1.13 | 9.97 ± 6.47 | 12.37 ± 7.07 | 19.77 ± 2.85 |
| **EuroSAT** | **Base** | Natural Acc | 18.07 ± 0.24 | 54.30 ± 17.51 | 49.03 ± 15.04 | 87.70 ± 1.57 |
| | | Adv Acc | 17.77 ± 0.19 | 15.90 ± 12.01 | 38.03 ± 8.41 | 51.80 ± 5.00 |
| | **New** | Natural Acc | 25.50 ± 4.98 | 26.73 ± 6.04 | 35.63 ± 3.13 | 32.80 ± 12.23 |
| | | Adv Acc | 19.97 ± 4.86 | 6.83 ± 5.77 | 19.47 ± 3.60 | 13.40 ± 10.38 |
| **OxfordPets** | **Base** | Natural Acc | 32.57 ± 37.86 | 38.97 ± 34.04 | 60.67 ± 39.22 | 87.37 ± 0.94 |
| | | Adv Acc | 12.27 ± 12.61 | 16.80 ± 19.18 | 31.80 ± 18.82 | 34.13 ± 8.01 |
| | **New** | Natural Acc | 32.30 ± 36.28 | 39.67 ± 34.97 | 57.90 ± 37.00 | 72.13 ± 6.21 |
| | | Adv Acc | 13.37 ± 13.53 | 17.50 ± 17.61 | 28.90 ± 16.69 | 26.07 ± 7.48 |
| **FGVCAircraft** | **Base** | Natural Acc | 2.30 ± 0.22 | 15.00 ± 7.03 | 9.93 ± 9.93 | 24.83 ± 0.12 |
| | | Adv Acc | 0.30 ± 0.16 | 6.63 ± 2.76 | 4.53 ± 3.07 | 8.00 ± 0.83 |
| | **New** | Natural Acc | 2.00 ± 0.00 | 9.97 ± 6.17 | 6.73 ± 6.22 | 15.83 ± 0.63 |
| | | Adv Acc | 2.00 ± 0.00 | 3.13 ± 1.13 | 2.50 ± 1.90 | 4.23 ± 0.74 |
| **Food101** | **Base** | Natural Acc | 2.27 ± 0.21 | 71.37 ± 0.05 | 71.40 ± 1.21 | 72.37 ± 1.44 |
| | | Adv Acc | 1.27 ± 0.61 | 27.90 ± 0.43 | 28.43 ± 0.34 | 27.57 ± 2.88 |
| | **New** | Natural Acc | 2.20 ± 0.36 | 68.93 ± 0.82 | 69.90 ± 0.28 | 68.20 ± 1.42 |
| | | Adv Acc | 1.00 ± 0.78 | 24.50 ± 0.22 | 24.60 ± 0.79 | 24.20 ± 2.70 |
| **Flowers102** | **Base** | Natural Acc | 50.43 ± 4.41 | 88.90 ± 0.49 | 56.53 ± 35.85 | 89.30 ± 0.41 |
| | | Adv Acc | 24.63 ± 2.80 | 62.80 ± 1.63 | 36.70 ± 25.23 | 65.50 ± 0.86 |
| | **New** | Natural Acc | 45.23 ± 2.69 | 49.90 ± 2.55 | 30.00 ± 18.02 | 45.67 ± 3.09 |
| | | Adv Acc | 15.77 ± 2.90 | 21.07 ± 1.86 | 11.63 ± 8.21 | 18.10 ± 0.54 |
| **StanfordCars** | **Base** | Natural Acc | 14.87 ± 19.89 | 56.47 ± 1.72 | 55.60 ± 0.54 | 53.97 ± 0.97 |
| | | Adv Acc | 2.77 ± 3.49 | 16.57 ± 0.29 | 16.97 ± 1.05 | 18.60 ± 0.64 |
| | **New** | Natural Acc | 15.53 ± 20.69 | 46.03 ± 1.89 | 46.00 ± 0.85 | 42.67 ± 1.08 |
| | | Adv Acc | 3.70 ± 3.96 | 12.10 ± 1.04 | 12.67 ± 0.57 | 14.10 ± 0.22 |
| **SUN397** | **Base** | Natural Acc | 60.20 ± 0.83 | 70.23 ± 0.31 | 70.57 ± 0.70 | 68.47 ± 0.56 |
| | | Adv Acc | 18.50 ± 0.71 | 33.87 ± 0.76 | 34.10 ± 0.73 | 34.63 ± 0.97 |
| | **New** | Natural Acc | 62.20 ± 0.73 | 63.57 ± 0.45 | 63.27 ± 0.76 | 61.47 ± 0.69 |
| | | Adv Acc | 21.10 ± 0.50 | 29.83 ± 0.76 | 29.40 ± 0.67 | 30.77 ± 0.97 |
| **UCF101** | **Base** | Natural Acc | 1.77 ± 0.52 | 72.77 ± 0.95 | 72.80 ± 0.64 | 70.37 ± 1.55 |
| | | Adv Acc | 1.73 ± 0.54 | 36.37 ± 0.19 | 36.77 ± 1.53 | 36.63 ± 0.48 |
| | **New** | Natural Acc | 2.47 ± 0.45 | 49.83 ± 3.07 | 50.70 ± 1.59 | 47.10 ± 3.11 |
| | | Adv Acc | 1.43 ± 0.82 | 20.13 ± 1.06 | 18.00 ± 1.77 | 18.30 ± 1.12 |

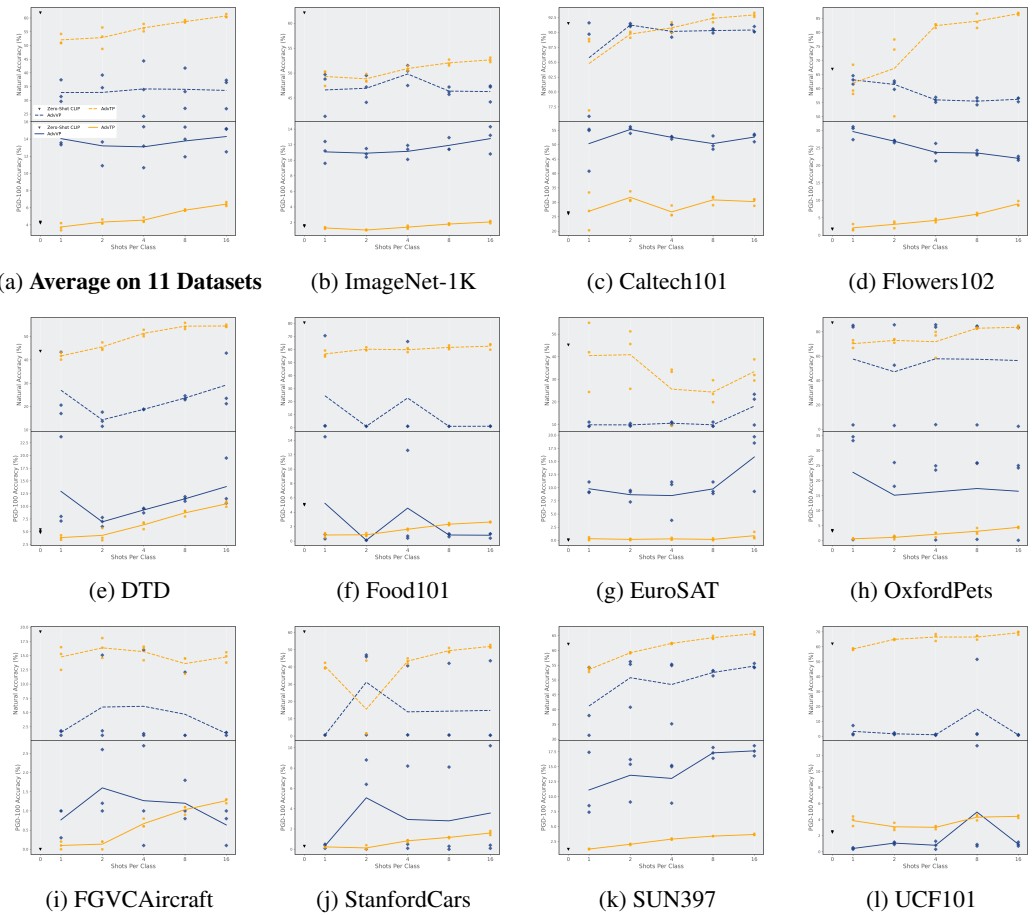

Figure 9: Accuracy (%) of adversarial few-shot learning on 11 datasets under uni-modal prompt AdvTP and AdvVP settings. The dots represent the result of each experiment and lines reveal the trend of the average results from three trials under each setting with respect to the shot numbers. In each subfigure, we report the natural accuracy (dashed line) in the upper half, and the robust accuracy (solid line) in the lower half.

# F Reproducibility

During the reviewing process, the source code is supplied anonymously as part of the supplementary materials. Additionally, upon the acceptance of the paper, this code will be publicly released.

# G Limitations

This paper introduces a framework that leverages the architecture of cross-modal prompts to enhance model robustness. This is achieved by adjusting the prompts to learn adversarial-correlated text supervision. However, prompt learning is merely a parameter-efficient strategy for model adaptation, and other parameter-based adaptation methods, such as full-finetuning, are not considered in this work. Furthermore, while our method has empirically shown that a comprehensive consideration of the connections and distinctions between natural and adversarial examples can better learn adversarial text supervision, a systematic theoretical analysis and proof remain elusive. We regard addressing these limitations as our future direction.

