# OpenReview forum: "Few-Shot Adversarial Prompt Learning on Vision-Language Models"
_NeurIPS.cc/2024/Conference — NeurIPS 2024 poster_

### Official Review · Reviewer_Rdwb · 2024-07-04

**Soundness:** 2
**Presentation:** 2
**Contribution:** 3
**Rating:** 6
**Confidence:** 2

**Summary:**

This paper proposes a few-shot adversarial training methodology for vision-language models to ensure robustness in downstream tasks of pretrained vision-language models. Specifically, instead of using traditional adversarial training methods, it adapts the TRADES loss, a prominent adversarial training approach, for vision-language models. Additionally, it leverages the structural characteristics of vision-language models by adaptively adjusting the weight of the regularization loss, enabling the model to effectively learn robustness in few-shot learning scenarios. As a result, the proposed methodology allows vision-language models to easily acquire robustness through few-shot learning, demonstrating high performance.

**Strengths:**

The methodology proposed by the authors effectively adapts the TRADES adversarial training method, originally used in other fields (vision), for training vision-language models. Additionally, they apply a uni-modal adversarial-aware mechanism to the TRADES loss. Through further analysis, the authors demonstrate that this mechanism enables the model to distinguish well between natural and adversarial examples, avoiding meaningless learning and promoting meaningful learning, which is highly suitable for few-shot learning. The authors point out that pretrained models often show insufficient performance on natural examples when learning adversarial examples for downstream tasks in few-shot learning scenarios. TRADES appears to be a highly suitable loss function for addressing this issue. Consequently, experiments show that the proposed method achieves superior performance in both clean and robust accuracy compared to existing methods.

**Weaknesses:**

- The motivation does not seem sufficient. Among the drawbacks of previous methods raised by the authors, it is not clearly conveyed how (1) and (2) are addressed and resolved later in the paper. Instead, it might have been better to emphasize in section 3.5 that existing methods or simply applying TRADES can lead to small perturbations that do not promote robust learning, which can be particularly disadvantageous in the constrained context of few-shot learning. For (3), if the issue is that the existing clean accuracy is compromised by training only with adversarial examples during the few-shot learning process, a baseline experiment analyzing the performance when both clean and adversarial examples are used together might have been more insightful.
- There is a lack of analysis regarding the proposed Uni-Modal Adversarial-Aware Mechanism. It is unclear why this method is implemented as a multiplicative factor to the KL divergence loss rather than adding a separate loss. The reasons behind the results shown in Figure 2 using this learning method are not clearly explained. While I understand that it is advantageous for learning if clean and adversarial examples are distinctly separated in the embedding space, I am concerned that if clean and adversarial examples are separated in this manner in the final trained model, it might actually reduce robustness.
- The analysis of the experiments is insufficient. Although the proposed methodology showed an increase in the overall average performance on the tested datasets, some datasets exhibited similar or even decreased performance compared to existing methods. A deeper consideration and explanation of these observations are necessary.

**Questions:**

Please provide answers to the mentioned points. Additionally,
- Were the adversarial examples in Figure 2 generated using PGD or were they generated by maximizing the KL divergence loss?
- Are there any comparative experimental results based on full fine-tuning?

**Limitations:**

Mentioned above

---

> ### Author Rebuttal · Authors · 2024-08-06
>
> ### Official Response to AnonReviewer Rdwb:
>
> We are genuinely grateful for your detailed assessment and valuable insights. Your constructive feedback has significantly contributed to the refinement and advancement of our study.
>
> **1. [Re Weakness 1: How we solve drawbacks of previous methods.]**
>
> Thanks for your feedback! We kindly argue that we have provided targeted solutions to each of the three drawbacks (**lines 113-130**) of the previous method. We illustrate them as follows:
>
> - For drawback (1), concerning the need for large datasets and lengthy prompts, we have implemented a few-shot setting and cross-modality prompt structures to reduce adaptation efforts.
>
> - For drawback (2), regarding hand-crafted text prompts lacking adversarial cues, we've integrated learnable text supervision that evolves end-to-end from adversarial visual examples.
>
> - For drawback (3), we agree that a baseline that uses both clean and adversarial examples will be insightful. We treat the TRADES-like loss design as such a baseline. From the comparison results (**lines 1 and 4 in Table 5**), we can find that the natural term brings about significant performance gain, which validates our analysis of drawback (3).
>
>   > ***Note:** We didn't use both clean and adversarial examples directly with two cross-entropy terms, as aligning both output features with one-hot labels through cross-entropy could cause conflicts. While for completeness, we included these results in **line 2 of Table 5** for ablation.
>
> **2. [Re Weakness 2: Analysis of the uni-modal auxiliary loss.]**
>
> Thanks for your feedback! We first clarify the contribution of the adversarial-aware mechanism as follows:
>
> - Although both natural and adversarial examples contain robust and predictive features consistent with human judgment, adversarial perturbations are predictive yet imperceptible features. We use an adversarial-aware mechanism to minimize reliance on such non-robust features.
>
> - The cross-modal consistency term $\mathcal{L}\_\text{KL}$ ealigns categorical features of natural and adversarial examples, aiding robust feature learning. The uni-modal adversarial-aware term $\mathcal{L}\_\text{cos}$ highlights differences, alerting the model to non-robust features and reducing sensitivity.
>
> - Practically, the cross-modal consistency term $\mathcal{L}\_\text{KL}$ seeks to balance two cosine similarities between text-image pairs. This can be regarded as aligning the angles between natural and adversarial text-image representations in the feature space. Two scenarios arise:
>
>   1.  The visual features are similar, being on the same side of the text feature.
>   2.  They differ but have the same angle with the text feature.
>
>   Our uni-modal adversarial aware mechanism maximizes the similarity between natural and adversarial visual features to ensure cross-modal consistency falls into the second scenario. Thus, $\mathcal{L}\_\text{cos}$ acts as an additional constraint for $\mathcal{L}\_\text{KL}$, reducing reliance on non-robust features.
>
> We then explain why the uni-modal loss is implemented as a multiplicative factor:
>
> - As illustrated above, the uni-modal term is an additive constraint for KL term, as it makes the cross-modal consistency fall as much as possible into the second case. Therefore, we should multiply $\mathcal{L}\_\text{cos}$ to $\mathcal{L}\_\text{KL}$.
> - Additionally, we typically follow the two-term formulation from TRADES. This aligns more closely with the trade-off between natural and adversarial samples in adversarial problems.
> - Comparative testing results in **Table Re.4** shows the three-term loss design slightly lowers performance. For better results and simpler hyper-parameter tuning, we prefer the two-term design.
>
> **3. [Re Weakness 3: Insufficient analysis of experimental results.]**
>
> We understand your concern. We answer it as follows:
>
> - Our method is tested across 11 diverse datasets, many of which pose challenges due to their deviation from general knowledge. Despite challenges, it generally performs well, achieving significant average gains.
> - Detailed analysis shows that our method excels in 9/12 downstream tasks in adversarial few-shot learning (**Figure 3**) except Caltech101, DTD, and FGVCAircraft, and in 9/12 tasks in adversarial cross-dataset transfer (**Table 8**) except DTD, EuroSAT, and Flowers102. These typically involve fine-grained classification with small inter-class differences among samples.
> - In these cases, our method might be less effective due to similar inter-class differences in adversarial examples. Using TeCoA loss to fit the label distribution of adversarial examples and allowing flexible learning through text prompts might better distinguish between classes.
>
> **4. [Re Question 1: Adversarial attack generation.]**
>
> Thanks for the feedback! Adversarial examples are generated using PGD in **Figure 2** as the results are visualized on the test set. All test-time attacks follows the same PGD settings for fair comparison.
>
> **5. [Re Question 2: Did not consider full fine-tuning setting.]**
>
> We understand your concern. We explain this as follows.
>
> - We prioritize the few-shot adversarial prompt setting as it's more relevant to real-world situations where large labeled datasets are costly. This approach suits VLMs particularly, which are resource-intensive in pre-training. We aim to reduce dependency on extensive downstream data by using semantic supervision to ease adaptation.
> - Additionally, we enhance robustness via prompt learning, an efficient method that doesn't need much time or resources. It uses the prior knowledge for quick adaptation to specific tasks. We focus on creating efficient prompt structures and objectives instead of depending on large datasets.
> - Lastly, our FAP shows better performance than SOTA, even with smaller datasets as shown in **Table 2**. This reveals VLMs' capacity in few-shot adversarial prompt, encouraging further exploration in such adaptations.

---

> ### Comment · Reviewer_Rdwb · 2024-08-10
>
> Thank you for the response. Most of my concerns and questions have been resolved. It seems that the drawbacks of previous works that served as motivation were broadly addressed by the methodology, making it difficult to distinguish them clearly. I appreciate the confirmation experiment on the uni-modal loss. While I wonder why TRADES has only now been applied to this task, it seems likely that this paper is the first to successfully adapt it to this task. This appears to be a commendable contribution. Accordingly, I raised my score.

---

> > ### Author Response · Authors · 2024-08-11
> >
> > Dear Reviewer Rdwb,
> >
> > Thank you for taking the time to review our paper. We appreciate your effort and are delighted to receive positive feedback from your comments！
> >
> > Best regards,
> >
> > Authors of #1695

---

### Official Review · Reviewer_M4S3 · 2024-07-09

**Soundness:** 3
**Presentation:** 3
**Contribution:** 3
**Rating:** 7
**Confidence:** 4

**Summary:**

This paper introduces a novel few-shot adversarial prompt framework for enhancing the adversarial robustness of vision-language models. The authors propose a method that achieves state-of-the-art zero-shot adversarial robustness using only 1% of training data, addressing limitations of existing approaches such as heavy adaptation costs and suboptimal text supervision. The key contribution is a new training objective that improves multi-modal feature consistency while differentiating uni-modal features between natural and adversarial examples.
The experiments rely on one multimodal model, but executes experiments on 12 datasets.
While the approach shows promise, potential limitations include questions about generalizability across different model architectures.
Overall, this paper presents an innovative approach to improving adversarial robustness in vision-language models with promising results.

**Strengths:**

- Novel approach: The paper introduces a few-shot adversarial prompt framework, which appears to be an innovative method for improving adversarial robustness in vision-language models.
- Efficiency: The approach achieves state-of-the-art zero-shot adversarial robustness using only 1% of training data, which is a significant improvement in data efficiency.
- Multi-modal consistency: The authors propose a novel training objective that enhances the consistency of multi-modal features while encouraging differentiated uni-modal features between natural and adversarial examples.
- Practical relevance: The method addresses real-world issues such as heavy adaptation costs and suboptimal text supervision in existing approaches.

**Weaknesses:**

- Generalizability: It's unclear how well this method generalizes across different vision-language model architectures. The authors should provide more information on the range of models tested.
- Comparative analysis: While the abstract mentions matching state-of-the-art performance, a more detailed comparison with existing methods would strengthen the paper.

Writing:
 - Line 74: a cross reference would ease reading.

**Questions:**

- Have you thought of how your method performs across different types of adversarial attacks?
- What are the limitations of your approach, and are there specific scenarios where it might not perform as well?

**Limitations:**

The authors have addressed some limitation in the appendix, directing towards more theoretical analysis of the method.


In my opinion, long-term robustness could be addressed as well: The paper could have addressed (at least in the future work) how the proposed method performs against evolving adversarial attacks such as adaptive attacks [1].

[1] https://proceedings.mlr.press/v162/croce22a.html

---

> ### Author Rebuttal · Authors · 2024-08-06
>
> ### Official Response to AnonReviewer M4S3:
>
> We sincerely thank you for your comprehensive examination of our paper and value the thoughtful feedback you have offered. Your helpful suggestions have played a crucial role in improving the overall quality of our research.
>
> **1. [Re Weakness 1: Generalizability.]**
>
> Thanks for your feedback! We answer your question as follows:
>
> - We kindly argue that most of the existing work on adversarial prompts, and even VLM prompts, is based on the CLIP model. This is because the dual encoder structure and contrastive multi-modal feature interactions are particularly suited for downstream classification and retrieval tasks. In this paper, we ensure the generalizability of our proposed method by testing it on various downstream tasks and alternative prompt structures **Appendix D.5**.
> - Furthermore, we step outside the existing framework and attempt to apply adversarial prompts on other VLMs. We have chosen the CLIP-RoBERTa **[1]** model, expending a learnable prompt of length 2 at the input sequences of each block in the image and text encoders. For simplicity, we adopt AdvVLP prompt structure, carrying out comparative experiments between baseline TeCoA loss and our learning objective. The results in **Table Re.1** show the significant improvements brought by our learning objective, especially in terms of generalization to new classes.
>
> **2. [Re Weakness 2: Comparative analysis.]**
>
> We sincerely appreciate your feedback! We explain this as follows.
>
> - The previous SOTA performance of adversarial prompt is achieved by AdvVP **[2]** which adapt token-level visual prompt (with size 200) with entire ImageNet 1K. In this work, we consider a few-shot setting to avoid heavy adaptation costs and add comparative prompt settings AdvTP, AdvVLP, and AdvMaPLe for completeness. We have summarized their respective design details in **Table 6** for reference.
> - Furthermore, we conduct a thorough review of recent advancements in adversarial prompts and identify a concurrent study AdvPT **[3]** at ECCV 2024 focusing on downstream adversarial robustness. Although this study also utilizes learnable text prompts and TRADE-format loss, our method emphasizes multi-modal prompt design and introduces a novel learning objective for few-shot learning. We validate the performance enhancements brought by our learning objective within this new prompt framework. Here we test it under our few-shot base-to-new generalization setting and report its average results across 8 datasets. We can conclude from **Table Re.2** that our proposed learning objective can also serve as a plug-in method to enhance the effectiveness of **[3]**, especially in terms of robust performance.
>
> **3. [Re Weakness 3: A cross reference in Line 74.]**
>
> Thanks for your positive suggestion! We kindly argue that we have added cross references at the end of each subsection of the method section for the potential discussion of corresponding paragraphs.  But we agree that a proper global cross reference to ablation studies and discussions will ease reading. We will add it in the final version. Thanks once again for the constructive suggestion!
>
> **4. [Re Question 1: Results under different adversarial attacks.]**
>
> We provide additional evaluation results in **Table Re.3** with 4 different attacks including the adaptive methods APGD-CE and APGD-DLR under the adversarial cross-dataset evaluation setting. Our observations indicate that our proposed FAP consistently outperforms the baselines.
>
> **5. [Re Question 2: Potential limitation under specific scenarios.]**
>
> Thanks for your feedback! Considering the inherent characteristics of efficient tuning, our proposed few-shot adversarial prompt (FAP) is designed to excel at rapidly adapting robustness from general knowledge to downstream tasks. Despite its advantages, the FAP is not a universal solution for all downstream problems. As far as we are concerned, there is one scenario where it may not be effective:
>
> - When downstream tasks significantly diverge from general knowledge, a natural distribution gap exists. This gap makes learning adversarial robustness using FAP from tasks with distribution shifts particularly challenging. For example, applying FAP on specialized datasets such as FGVCAircraft (fine-grained aircraft images) and DTD (texture dataset) does not yield significant improvements in adversarial robustness.
>
> Given that prior knowledge cannot effectively generalize to the aforementioned tasks, we conjecture that full-finetuning might be more suitable than prompt learning for adversarial adaptation in such settings.
>
> **6. [Re Limitation 1: Future work on evolving adversarial attacks.]**
>
> We appreciate your constructive suggestions! We had the following discussions:
>
> - As you mentioned, our proposed method is likely to be effective against evolving adversarial attacks when paired with adaptive test-time defenses. This synergy arises because both strategies share a common goal to rapidly adapt to input changes and enhance adversarial robustness.
>
> - Within the context of prompt architecture, we think the test-time prompt framework **[4]** will be more suitable to be compatible with adaptive test-time adversarial defense, as it tunes adaptive prompts on the fly with a single test sample. In other words, such a combination sheds light on adaptive test-time adversarial defense for VLMs, which seems to be a more concise yet powerful defense method for the long-term robustness of VLMs.
>
>
>
> [1] Cherti, Mehdi, et al. "Reproducible scaling laws for contrastive language-image learning." CVPR, 2023.
>
> [2] Mao, Chengzhi, et al. "Understanding zero-shot adversarial robustness for large-scale models." ICLR, 2023.
>
> [3] Zhang, Jiaming, et al. "Adversarial prompt tuning for vision-language models." ECCV, 2024.
>
> [4] Shu, Manli, et al. "Test-time prompt tuning for zero-shot generalization in vision-language models." NeurIPS, 2022.

---

> > ### Comment · Reviewer_M4S3 · 2024-08-13
> >
> > Thanks for your updates. You have clarified my questions. Ad 4) Please, be careful when using the word “adaptive”, because it does not mean that the APGD attack is not adaptive at test-time. Your results on the C&W attack are impressive.

---

> > > ### Author Response · Authors · 2024-08-13
> > >
> > > Dear reviewer M4S3:
> > >
> > > Thanks for your feedback and claiming that we have clarified your questions and the results on the C&W attack are impressive. We will take your advice and reflect it in the final version of this paper.
> > >
> > > Warm regards,
> > >
> > > Author of #1695

---

### Official Review · Reviewer_Lycp · 2024-07-13

**Soundness:** 3
**Presentation:** 3
**Contribution:** 3
**Rating:** 6
**Confidence:** 4

**Summary:**

Adversarial prompt learning on vision-language models has traditionally focused on aligning text with corresponding images to ensure coherence and contextual accuracy. This paper extends this approach by making the image features of natural and adversarial examples distinct while still aligning them with the relevant text descriptions. This novel contribution enhances the model's robustness against adversarial attacks. The authors introduce a new framework that leverages adversarial text supervision to improve cross-modal adversarial alignment. This framework allows for the learning of adversarial prompts, significantly boosting the model’s ability to handle adversarial examples. Remarkably, the proposed method achieves state-of-the-art zero-shot adversarial robustness while utilizing only 1% of the training data, demonstrating both efficiency and effectiveness in enhancing model performance.

**Strengths:**

Novelty: The paper introduces a training objective that enhances the consistency of multi-modal features while encouraging differentiated uni-modal features between natural and adversarial examples.

Clarity: The paper is clearly written, with methodologies and results presented in an accessible and comprehensible manner.

Significance: The proposed approach is compared with three baseline methods and demonstrates significant improvements in robust accuracy across 11 different datasets.

**Weaknesses:**

Depth of Analysis: The paper introduces a novel training objective that is both interesting and unexpected. However, there is a need for a deeper analysis and explanation of why this proposed method is effective. Understanding the underlying mechanisms and reasons for its success would provide valuable insights and strengthen the paper's contributions.

**Questions:**

When would this few-shot setup particular useful? If we already have LAION and other big dataset?

**Limitations:**

No.

---

> ### Author Rebuttal · Authors · 2024-08-06
>
> ### Official Response to AnonReviewer Lycp:
>
> We sincerely thank you for your careful reading of our paper and appreciate the valuable feedback in your comments. The insightful and constructive suggestions have enabled us to effectively improve our work.
>
> **1. [Re Weakness 1: Depth of Analysis for training objective.]**
>
> Thanks for your feedback. Our training objective generally exhibits adversarial regularization formulation **[1]**, with a natural term ensuring natural generalization and an adversarial term enforcing output consistency on adversarial examples. In this framework, we creatively decompose the adversarial term into a cross-modal consistency term and a uni-modal adversarial aware term for effective few-shot adversarial prompting on VLMs. Here, we first summarize the intuition of such decomposition and differentiated processing between features of different modalities, then present a deep analysis of its success.
>
> - Intuition. Although both natural and adversarial examples contain robust and predictive features consistent with human judgment, adversarial perturbations are predictive yet imperceptible features. Adversarial vulnerability is a direct result of the sensitivity to such well-generalizing yet non-robust features **[2]**. Therefore, an effective prompt training objective must reasonably deduce the relationships and differences between natural and adversarial samples using a limited number of examples.
>
> - The cross-modal consistency term $\mathcal{L}\_\text{KL}$ ensures the decisive categorical features of natural and adversarial examples are consistent, aiding in robust feature learning. In contrast, the uni-modal adversarial-aware term $\mathcal{L}\_\text{cos}$ highlights differences between natural and adversarial images, alerting the model to non-robust features and preventing sensitivity to them.
>
> - Practically, the cross-modal consistency term $\mathcal{L}\_\text{KL}$ seeks to balance two cosine similarities between text-image pairs: $\operatorname{cos}(\mathbf{z}^{(I,\boldsymbol{P}\_{\boldsymbol{v}})},\mathbf{z}^{(t\_i,\boldsymbol{P}\_{\boldsymbol{t}})})$ and $\operatorname{cos}(\tilde{\mathbf{z}}^{(I,\boldsymbol{P}\_{\boldsymbol{v}})},\mathbf{z}^{(t\_i,\boldsymbol{P}\_{\boldsymbol{t}})})$. This can be regarded as aligning the angles between natural and adversarial text-image representations in the feature space. There are two possible scenarios:
>
>   1.  The visual features are similar, being on the same side as the text feature.
>   2.  They are on opposite sides, differing from each other but having the same angle with the text feature.
>
>   Our proposed uni-modal adversarial aware mechanism maximizes another similarity $\operatorname{cos}\left(\mathbf{z}^{(I,\boldsymbol{P}\_{\boldsymbol{v}})},\tilde{\mathbf{z}}^{(I,\boldsymbol{P}\_{\boldsymbol{v}})}\right)$ between natural and adversarial visual features to ensure that the cross-modal consistency primarily falls into the second scenario. As a result, $\mathcal{L}\_\text{cos}$ act as an additional constraint for $\mathcal{L}\_\text{KL}$, excluding conditions where the model relies on the non-robust features to minimize the loss.
>
> **2. [Re Question 1: Usefulness of the few-shot setup.]**
>
> We address this from the following aspects:
>
> - Large datasets are particularly useful for pre-training powerful foundation models such as CLIP, BERT, and ViT with general knowledge. However, few-shot learning is typically used in downstream adaptation, where we tune the pre-trained model with a small number of task-specific samples to enhance its generalization capability on the downstream task. From this perspective, large datasets can facilitate few-shot learning in that pre-trained models from large datasets tend to carry more diverse prior knowledge, making the adaptation to downstream tasks less data-hungry.
>
> - Furthermore, few-shot learning is especially useful for the following advantages:
>
>   1. **Data efficiency.** Many downstream tasks have only limited data available. A typical example is facial recognition tasks, which have only one-shot data as reference but need to perform recognition across different lighting conditions, facial expressions, and backgrounds.
>
>   2. **Rapid adaptation.** In many cases, there might be a large amount of similar downstream tasks. Few-shot learning enables rapid adaptation for each downstream task, whereas fine-tuning on a large dataset is too cumbersome. A typical example is a customized chatbot.
>
>   3. **Computational efficiency.** The computational efficiency of few-shot learning makes it promising for downstream tasks with high computational complexity. For example, in adversarial robustness tasks, the iterative process of adversarial attack and defense significantly increases the required time compared to natural training. Therefore, attempting few-shot adversarial learning from a pre-trained model with initial robustness is also highly practical.
>
> [1] Zhang, Hongyang, et al. "Theoretically principled trade-off between robustness and accuracy." ICML, 2019.
>
> [2] Ilyas, A., Santurkar, S., Tsipras, D., Engstrom, L., Tran, B., and Madry, A. Adversarial examples are not bugs, they are features. NeurIPS, 2019.

---

> > ### Comment · Reviewer_Lycp · 2024-08-07
> > **Thank you for the reply**
> >
> > My concerns are answered.

---

> > > ### Author Response · Authors · 2024-08-08
> > >
> > > Thank you for your reply. Your comments really help improve this work.

---

### Official Review · Reviewer_KV22 · 2024-07-20

**Soundness:** 2
**Presentation:** 2
**Contribution:** 2
**Rating:** 5
**Confidence:** 3

**Summary:**

This paper addresses adversarial robustness for image classification with VLMs (e.g. CLIP model). To this end, the authors proposed the FAP framework to adapt VLM models by learning prompt tokens in a few-shot manner with adversarial examples. The loss function is proposed to promote accuracy on clean images, to align the text-image embedding interaction across clean and adversarial images and to align the image embedding similarity across clean and adversarial images. Experiments are carried out on image classification tasks, with the few-shot setup on multiple datasets, and the proposed method achieves good results on both base and novel datasets.

**Strengths:**

* The results are indeed better when there are not many training examples.
* Extensive discussions are provided in the appendix.

**Weaknesses:**

* The framework architecture is not easy to understand without reading the MaPLe paper. For example, is $P_v$ in line 150 shared shared or separate across different encoder layers? And the modification of $h$'s direction is not clear without reading the appendix. The presentation should be improved and more self-contained.
* From Table 1, it seems that most of the gains are coming from the architecture of MaPLe. For the proposed loss in equation 7, the components of them are not carefully ablated.

Nitpicking:
Line 108: $\theta_T$ -> $\theta_I$

**Questions:**

In Eq 3 and 4, $\bf{t}$ seems to be a list of strings/text prompts (following line 83), how is $cos$ computed on strings? Or are we missing some $z$ here?
In Appendix A Algorithm1, are $P_v$ and $P_t$ separate or one is dependent on another? It's not very clear in that codeblock.

**Limitations:**

Limitations are adequately addressed in the appendix.

---

> ### Author Rebuttal · Authors · 2024-08-06
>
> ### Official Response to AnonReviewer KV22:
>
> We deeply appreciate your thorough review of our manuscript and are grateful for the insightful feedback you provided. Your constructive comments have been instrumental in enhancing the quality of our work.
>
> **1. [Re Weakness 1: Concerns about the comprehensibility of the framework.]**
>
> Thanks for your positive suggestions! We would like to first answer the questions and then explain the framework design of our paper.
>
> - $P_v$ is separate across different encoder layers, which is a standard configuration of deep prompt tuning structure **[1]**. Besides, the direction of prompt projection $h$ is analyzed in **Appendix D.2** and we place a cross reference in **Section 3.1** to ease reading.
> - As you mentioned, MaPLe serves as the structural foundation for prompt learning in this work but we make both architectural and learning objective modifications for adversarial robustness adaptation. We suppose the preliminary content about prompt learning and adversarial prompts in **Section 2**, along with the overall framework in **Figure 1**, should help readers understand the basic scheme of prompt learning and the method we proposed.
> - We understand your concern. After carefully re-evaluating our paper, we agree that presenting the results appropriately will enhance the paper's quality and make it more self-contained. Including the analytical experiments on our architecture (from **Appendix D.2**) and learning objectives improvements (in **Table 5**) in the main text will better help readers understand our method. We will move the content from **Appendix D.2** to the main text in the final version. Thank you once again for the constructive advice!
>
> **2. [Re Weakness 2: Performance gain over AdvMaPLe and loss ablation.]**
>
> Thanks for your feedback! We address your concern from the following three aspects:
>
> - From our initial experiment, we find that AdvMaPLe exhibits powerful performance in few-shot adversarial prompt learning owing to its better-aligned multi-modal structure. However, its instability in both structure (**Appendix D.2**) and learning objectives (**Appendix D.3**) makes it suboptimal.
> - We kindly argue that the improvement brought about by our method is significant in **Table 1**, which obtains [+10.14%, +7.36%, +3.40%, +1.61%] performance gain on [base nat acc, base adv acc, new nat acc, new adv acc]. Additionally, our method is tailored for few-shot adversarial prompts, as it reveals superior stability and fits other prompt frameworks (**Appendix D.5**) with evident performance gain as well.
> - We have actually presented a comprehensive loss function ablation in **Table 5**, which demonstrates the effectiveness of our proposed learning objective. Further, we validate the effectiveness of our learning objective in the few-shot adversarial prompt task on other prompt frameworks in **Appendix D.5** and alternative VLMs in **Table Re.1**.
>
> **3. [Re: weakness 3: Typo error in Line 108.]**
>
> Thank you! We will correct it in the final version.
>
> **4. [Re question 1: Questions related to prompt design.]**
>
> We understand your concern. We clarify the two questions about prompt design as follows:
>
> - Here, $\textbf{t}$ represents a list of text prompts (**dim:** [num_classes, embed_dim]), as indicated by its bold formatting, while $I$ represents a single image embedding (**dim:** [1, embed_dim]), as it is not bolded. The term `cos` does not refer to a single cosine similarity value but rather to a cosine similarity distribution, as outlined in the original CLIP paper **[2]**. Specifically, we calculate the cosine similarity between one image feature (**dim:** [1, embed_dim]) and all text features (**dim:** [num_classes, embed_dim]) from every class to obtain this distribution (**dim:** [1, num_classes]). This process is illustrated in **lines 174-176**. We will include these explanations in the final version of the paper to ensure clarity.
> - $P_t$ and $P_v$ are relevant in our method. As described in **line 152-154**, $P_t$ is obtained from $P_v$ through linear projection $h$, as $P_t=h\left(P_v\right)$. We realize that the brief description in **Algorithm 1** might confuse, so we will explicitly clarify the projection relationship between image and text prompts in **Algorithm 1** in the final version. Thanks once again for your constructive suggestions.
>
>
>
> [1] Jia, Menglin, et al. "Visual prompt tuning." ECCV, 2022.
>
> [2] Radford, Alec, et al. "Learning transferable visual models from natural language supervision." ICML, 2021.

---

> > ### Author Response · Authors · 2024-08-12
> >
> > Dear Reviewer KV22,
> >
> > Thanks for your thorough review of our paper. We have made every effort to address the concerns you raised. As the deadline for discussions between reviewers and authors is approaching, we would like to confirm whether there are any explanations or descriptions in our response that remain unclear. We are prepared to provide further clarification if needed.
> >
> > Warm regards,
> >
> > Authors of #1695

---

> > > ### Comment · Reviewer_KV22 · 2024-08-13
> > >
> > > thanks for addressing my questions and helping to improve the presentation of the paper. I updated the rating to Borderline accept given the authors' response.

---

> > > > ### Author Response · Authors · 2024-08-13
> > > >
> > > > Dear Reviewer KV22，
> > > >
> > > > Thank you for your time and effort in reviewing our paper. We are grateful for your feedback and pleased to hear your positive remarks!
> > > >
> > > > Best regards,
> > > >
> > > > Authors of #1695

---

### Author Rebuttal · Authors · 2024-08-06

### General response

We appreciate the reviewers’ insightful comments and constructive feedback on our manuscript. We are pleased to receive positive feedback from most of the reviewers. Furthermore, we are delighted to learn that the reviewers found the idea of the proposed method to be novel and effective (all reviewers), the experiments to be overall convincing and impressive (Reviewers KV22, Lycp, and M4S3), and the concept illustration/analysis is clear and informative (Reviewers KV22, Lycp, and M4R3). Based on the comments, we provide a general response to the points raised by multiple reviewers and individual responses below to address each reviewer’s concerns.

(1) Regarding the questions about the experiments, we have taken the following actions:

- For Reviewer M4S3, we have provided experimental results about different vision-language models.

- For Reviewer M4S3, we have supplemented comparative experiments on existing methods.

- For Reviewer M4S3, we have added experiments with different attack methodologies.

- For Reviewer Rdwb, we have provided experiments with different design choices for the uni-modal adversarial aware term.

  > ***Note: We have provided the experimental results mentioned above in the PDF, numbered from Table Re.1 to Table Re.4.**

(2) We have addressed the questions about the proposed problem, the idea, and technical details as follows:

- For Reviewer KV22, we have addressed the questions about framework comprehensibility, loss ablation, and prompt design.
- For Reviewer Lycp, we have addressed the concerns about the methodology explanation and usefulness of the few-shot setup.
- For Reviewer M4S3, we have solved the questions about the generalizability, comparative analysis, adversarial attack diversity, additional limitation, and further application on test-time adaptive attacks.
- For Reviewer Rdwb, we have answered the questions about the motivation, adversarial-aware mechanism analysis, and further experimental analysis.

(3) For the valuable suggestions of all reviewers regarding the presentation and organization of this paper, we will take them and finish modifications accordingly in the final version of this paper.

We sincerely thank all the reviewers for their constructive suggestions. Please feel free to let us know if further details/explanations would be helpful.

Yours sincerely,

Authors of #1695

---

### Decision · Program_Chairs · 2024-09-25

**Decision:**

Accept (poster)

**Comment:**

This paper proposes a few-shot adversarial prompt framework aimed at improving the adversarial robustness of vision-language models (VLMs). Overall, all reviewers enjoyed reading this paper and found the proposed method is novel and the provided empirical results are strong. But meanwhile, several concerns are raised, including some technique aspects need to be further clarified, and additional ablations are needed.

The authors' rebuttal successfully addresses most of these concerns, and all reviewers unanimously agree to accept this paper.